

# CMIP5 model selection for ISMIP6 ice sheet model forcing: Greenland and Antarctica

Alice Barthel[1], Cécile Agosta[2], Christopher M. Little[3], Tore Hattermann[4,5], Nicolas C. Jourdain[6], Heiko Goelzer[7,8], Sophie Nowicki[9], Helene Seroussi[10], Fiammetta Straneo[11], and Thomas J. Bracegirdle[12]

[1]Los Alamos National Laboratory, Los Alamos, NM, USA
[2]Laboratoire des Sciences du Climat et de l'Environnement, LSCE-IPSL, CEA-CNRS-UVSQ, Université Paris-Saclay, F-91198 Gif-sur-Yvette, France
[3]Atmospheric and Environmental Research, Inc., Lexington, Massachusetts, USA
[4]Norwegian Polar Institute, Tromsø, Norway
[5]Energy and Climate Group, Department of Physics and Technology, The Arctic University – University of Tromsø, Norway
[6]Univ. Grenoble Alpes/CNRS/IRD/G-INP, IGE, Grenoble, France
[7]Institute for Marine and Atmospheric research Utrecht, Utrecht University, Utrecht, the Netherlands
[8]Laboratoire de Glaciologie, Université Libre de Bruxelles, Brussels, Belgium
[9]NASA GSFC, Cryospheric Sciences Branch, Greenbelt, USA
[10]Jet Propulsion Laboratory, California Institute of Technology, Pasadena, CA, USA
[11]Scripps Institution of Oceanography, University of California San Diego, La Jolla, CA, USA
[12]British Antarctic Survey, Cambridge, UK

**Correspondence:** Alice Barthel (abarthel@lanl.gov)

**Abstract.** The ice sheet model intercomparison project for CMIP6 (ISMIP6) effort brings together the ice sheet and climate modeling communities to gain understanding of the ice sheet contribution to sea level rise. ISMIP6 conducts standalone ice sheet experiments that use space- and time-varying forcing derived from atmosphere-ocean coupled global climate models (AOGCMs) to reflect plausible trajectories for climate projections. The goal of this study is to recommend a sub-set of CMIP5

5    AOGCMs (3 core + 3 targeted) to produce forcing for ISMIP6 stand-alone ice sheet simulations, based on: i) their representation of current climate near Antarctica and Greenland relative to observations, and (ii) their ability to sample a diversity of projected atmosphere and ocean changes over the 21st century. The selection is performed separately for Greenland and Antarctica. Model evaluation over the historical period focuses on variables used to generate ice sheet forcing. For stage (i), we combine metrics of atmosphere and surface ocean state (annual- and seasonal-mean variables over large spatial domains) with

10   metrics of time-mean sub-surface ocean temperature biases averaged over sectors of the continental shelf. For stage (ii), we maximize the diversity of climate projections among the best performing models. Model selection is also constrained by technical constraints, such as availability of required data from RCP2.6 and RCP8.5 projections. The selected top 3 CMIP5 climate models are CCSM4, MIROC-ESM-CHEM, and NorESM1-M for Antarctica, and HadGEM2-ES, MIROC5 and NorESM1-M for Greenland. This model selection was designed specifically for ISMIP6, but can be adapted for other applications.



# 1 Introduction and objectives

The Greenland and Antarctic ice sheets represent the largest and most uncertain contribution to multidecadal to millenial timescale sea-level rise. During the last three decades, satellite observation captured rapid mass loss from both ice sheets (Khan et al., 2014; Mouginot et al., 2014; Zwally et al., 2011; Velicogna, 2009). Both atmospheric and oceanic changes have been identified as drivers of observed mass loss, although regional mechanisms vary. For example, rising air temperatures over Greenland lead to increased surface melt causing direct mass loss (Trusel et al., 2018; Fettweis et al., 2017). Enhanced surface melt water production also destabilizes the margins of the ice sheet (van den Broeke, 2005; Banwell et al., 2013) and lubricates the ice flow at the bed (Kendrick et al., 2018; Andrews et al., 2014). Ocean interactions with the ice sheet occur in Greenland fjords, where a combination of on-shore ocean heat transport, estuarine-type circulation, subglacial meltwater runoff, and calving processes influence glacier terminus position and ice discharge (Straneo and Cenedese, 2015). In Antarctica, most of the ice sheet's mass loss is mediated through floating ice shelves. Melting at the ice shelf underside, which affects ice flow dynamics, is mainly controlled by the extent to which ocean dynamics along the continental margin allow intrusion of offshore ocean heat into the ice-shelf cavities, leading to distinct regimes operating in 'warm' vs. 'cold' continental shelf regions (e.g., Dinniman et al., 2016; Thompson et al., 2018). Rising air temperatures and associated surface melting are thought to be responsible for the collapse of ice shelves around the Antarctic Peninsula (Domack et al., 2005) and subsequent speed up of grounded ice flow (Rignot et al., 2004), while surface melting is currently limited in most other parts of the continent (e.g., Trusel et al., 2013). In the future, increased water vapor transport in a warmer atmosphere may lead to increased surface accumulation in Antarctica (Frieler et al., 2015; Palerme et al., 2017) together with increased melting over Greenland (Franco et al., 2013) and the Antarctic ice shelves (Trusel et al., 2015). Besides this general pattern, the spatial distribution and magnitudes of atmospheric and oceanic mass balance contributions vary greatly over both ice sheets and depend on synoptic-scale climate variability and physical processes at regional and smaller scales.

The ice sheet model intercomparison project for CMIP6 (ISMIP6) effort brings together the ice sheet and climate modeling communities to gain understanding of the ice sheet contribution to sea level rise (Nowicki et al., 2016). (Due to the delay in CMIP6 AOGCM dataset release, ISMIP6 revised the protocol described in Nowicki et al. (2016) to utilize climate forcing from the CMIP5 dataset (Nowicki et al., in prep).) ISMIP6 conducts standalone ice sheet experiments that use space- and time-varying forcing derived from atmosphere-ocean coupled global climate models (AOGCMs) to reflect plausible trajectories for climate projections, building on earlier coordinated experiments which applied ad-hoc boundary conditions either constant in time, or imposed as an abrupt perturbation (Pattyn et al., 2013; Bindschadler et al., 2013; Levermann et al., 2014). However, this effort requires translating AOGCM output to forcing for ice sheet models, posing several challenges. First, climate models from the Coupled Model Intercomparison Project (CMIP) have a horizontal resolution too coarse to accurately represent sharp ice-sheet topographic gradients that impact surface climate over the ice sheet (e.g. melt, wind, precipitation). Ocean components cannot represent narrow fjords connecting the deep ocean and tidewater glaciers around Greenland (Straneo et al., 2012), or the ocean eddies involved in poleward heat transport across continental shelves (Stewart et al., 2018), or ocean circulation



beneath ice shelves. Second, AOGCMs poorly represent polar-specific processes that have a major impact on the ice sheet
surface climate (e.g. snowpack evolution, cloud and boundary-layer processes).

These limitations can be addressed by using regional climate models adapted for the polar regions. On the atmosphere side, polar-oriented regional climate models (RCMs) have proved to provide more realistic surface climate than direct AOGCM outputs for both the Greenland ice sheet (Noël et al., 2018; Fettweis et al., 2013) and the Antarctic ice sheet (van Wessem et al., 2018; Agosta et al., 2019). On the ocean side, a number of models have recently added the capability to represent ice shelf
cavities and ice/ocean interactions (e.g., Dinniman et al., 2016). However, ocean simulations are still challenged to provide non-biased solutions from a pan-ice sheet perspective, and remain computationally expensive, which probably explains the small number of existing projections of ice-shelf basal melting (Timmermann and Goeller, 2017; Naughten et al., 2018). Thus, the ISMIP6 steering committee has proposed the following strategy to convert AOGCM outputs into ice sheet forcing: surface forcing is provided by AOGCMs dynamically downscaled with a polar-oriented atmospheric RCM (Fettweis et al., 2017), while
ocean forcing is computed by interpolating AOGCMs ocean temperature onto the continental shelf and by parameterizing ice shelf melt or retreat rates, as detailed in Slater et al. (2019) and Jourdain et al. (in prep).

The goal of this study is to recommend a sub-set of CMIP AOGCMs to produce forcing for ISMIP6 standalone ice sheet simulations. This ensemble of AOGCMs aims to capture (i) plausible climate near Antarctica and Greenland over the historical period, and (ii) a diversity of atmosphere and ocean warming rates over the 21st century. For evaluating AOGCMs we focus on
variables that are inputs of the downscaling methods defined to generate ice sheet forcing. Although it is technically possible to select different AOGCMs for atmosphere and ocean forcing, we choose to use the same climate models across both realms due to their inter-dependence in projections (e.g., Krinner et al., 2014; Bracegirdle et al., 2018). We thus perform a combined assessment of both the atmosphere and ocean component of AOGCMs.

This paper describes the process utilized to select 6 AOGCMs to provide forcing for each ice sheet. This evaluation combines
observational/reanalysis data, metrics from existing studies and data produced specifically for this study. The methodology to combine distinct metrics for the ocean and atmosphere into a single ranking is detailed in Section 2. The models are selected independently for the Antarctic (Section 3) and the Greenland (Section 4) ice sheets. Finally, we present some of the assumptions underlying our analysis, and discuss perspectives for future research in Section 5.

## 2  Data and methods

### 2.1  General methodology

We analyze monthly output from 33 climate models of the CMIP5 ensemble, listed in Table 1. The ISMIP6 standalone experiment requires 3 coupled climate models to derive forcing fields for their core experiments (core), plus 3 additional models to extend the ensemble to a total of 6 models (targeted). To select the models, we first rank them according to their performance in reproducing observations over the 1979-2005 historical period (historical metrics, defined in Section 2.2). In a second step,
we define climate change metrics over the 21st century (21C) under the RCP8.5 scenario (Section 2.3.1) in order to select a set





of models that represents a diversity of 21C changes (Section 2.3.2). This two-step process is performed independently for the Antarctic and Greenland ice sheets.

The top 3 (core) models are those maximizing the diversity of climate change (Section 2.3.2, $n = 3$) among those fitting the following criteria:

1. the model must provide 6-hourly wind outputs, to be able to run an atmospheric regional climate model (18 models);

2. the model output must include required data fields under both the RCP2.6 and RCP8.5 scenario projections, following the revised ISMIP6 protocol (Nowicki et al., in prep) (25 models);

3. the model must rank in the top half of the 33-model ensemble with regard to the historical metrics defined in Section 2.2 (17 models, Fig. 2a and Fig. 5a);

4. the model must not have any single climate change metric defined in Section 2.3.1 above 2 interquartile range (IQR, equal to quantile 75 % minus quantile 25 %) from the multi-model median projection (Fig. 4a and Fig. 7a).

For the additional 3 models (targeted), criteria used for the top 3 are relaxed, now including models without sub-daily frequencies for Antarctica, and including models with projected 21C changes above 2 IQR of the multi-model median. The models are selected to maximize the diversity of climate change across the top 6-model ensemble (Section 2.3.2, $n = 6$). As
the selection method maximizing diversity tends to favor models with extreme values, we impose one model (within the top 6) which features 21C climate changes in the median range of the ensemble.

## 2.2 Historical metrics

### 2.2.1 Atmosphere and surface ocean metrics

For the atmosphere and surface ocean, we consider variables that have an impact on RCM-modeled surface mass balance and
for which reanalyses are reliable, following Agosta et al. (2015). All model outputs are bi-linearly interpolated onto a common regular longitude–latitude grid (1.5°x1.5°). For each variable that retains spatial information (described in the following paragraph), we calculate the spatial root mean square error (RMSE) for annual- or seasonal-mean values over 1980-2004 (25 years). We take the European Centre for Medium-Range Weather Forecasts "Interim" re-analysis (ERA-Interim, 1979–present; Dee et al. (2011)) as a reference, since differences between reanalyses are much smaller than climate model biases (Agosta
et al., 2015) and ERA-Interim was assessed to be the most reliable contemporary global reanalysis over Antarctica (Bromwich et al., 2011; Bracegirdle and Marshall, 2012; Huai et al., 2019; Gossart et al., 2019).

For Antarctica, we evaluate air temperature at 850 hPa (average of summer and winter RMSE), annual precipitable water, and annual sea-level pressure, together with summer sea surface temperature and winter sea ice extent, for the domain extending south of 40°S over the ocean (Fig. 1a). In addition to spatially-resolved variables, we include a metric of the historical CMIP5
vs ERA-Interim bias in westerly jet strength, calculated as the maximum in annual mean zonal mean 850 hPa zonal wind between 10°S and 75°S (in m s$^{-1}$), with comparisons made over time-slice means of the overlapping 1979-2005 period, as in Bracegirdle et al. (2018).



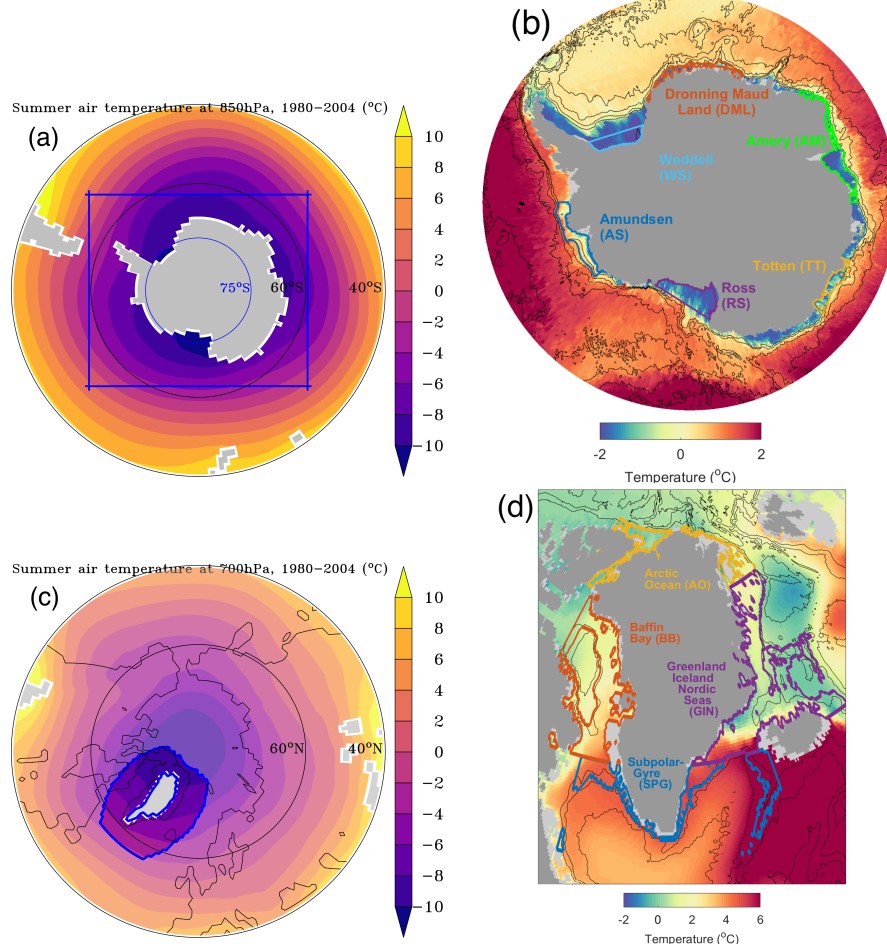

**Figure 1.** Atmosphere and ocean regions defined for metric computation. (a) For Antarctic atmosphere and surface ocean metrics, we considered the domain south of 40°S over ocean (color shading). The blue box shows standard lateral boundaries for regional climate models. Color shading is ERA-Interim summer air temperature at 850 hPa over 1980-2004. (b) For Antarctic ocean metrics, we considered 6 ocean sectors shallower than 1500 m. Color shading shows the depth-integrated temperature of our reference historical climatology. (c) For Greenland atmosphere metrics, we considered the domain inside the usual boundaries of MAR simulations in that region, i.e. inside the blue box, except where ice sheet topography is above 2000 m a.s.l. (bright color shading). Color shading is ERA-Interim summer air temperature at 700 hPa over 1980-2004. (d) For Greenland ocean metrics, we considered the 4 sectors shown with different colored outlines. Color shading shows the depth-integrated (200 to 500 m) temperature of our reference historical climatology.

For Greenland, we evaluate air temperature at 700 hPa (average of summer and winter RMSE), annual precipitable water, and annual geopotential height at 500 hPa, inside the RCM domain and where the Greenland ice sheet is below 2000 m a.s.l. (bright shaded color in Fig. 1c). In this small domain, sea surface conditions do not impact RCM results (Noël et al., 2014).





### 2.2.2 Sub-surface ocean metrics

The ISMIP6 standalone ice sheet oceanic forcing is derived from "far-field" salinity and potential temperature (Slater et al., 2019; Jourdain, in prep). Consistent with this approach, our evaluation of sub-surface ocean properties is performed on regionally-averaged CMIP5 temperatures. Since the oceans around Greenland and Antarctica are characterized by different geographic and dynamic regimes in observations (e.g., Straneo et al., 2012; Schmidtko et al., 2014; Thompson et al., 2018) and models (Yin et al., 2011; Little and Urban, 2016; Levermann et al., 2014), individual metrics are obtained for several sub-regions surrounding both ice sheets (Fig. 1b,d).

For this purpose, 1989-2009 time-mean ocean temperatures from each CMIP5 model are interpolated onto a common tripolar ORCA025 grid (Ferry et al., 2012), which has a quasi-isotropic resolution corresponding to 0.25 degrees in latitude, and 75 vertical layers with a thickness ranging from 1 m at the surface to 200 m at the bottom. We use a conservative 3d interpolation; if some parts of the ORCA025 grid are not covered by the CMIP grid, we extrapolate from the closest neighbour (horizontally above sills, then vertically to fill troughs behind sills). The regridding tools are made available on https://github.com/nicojourdain/SCRIPTS_CMIP5_ANOM_NOW (last access: 29 july 2019, Dutheil et al., 2019). Regionally averaged coastal ocean temperatures are then computed in six sectors around the Antarctic continent (Fig. 1b), which capture different continental shelf and melting regimes. A maximum bottom depth criterion of 1500 m is used, together with an explicit limit for the northern boundaries in the large embayments in the Ross and Weddell Seas, to select ORCA25 ocean cells that are located on the continental shelf near the coast. For Greenland, the ocean has been separated in four connected regions based on the major hydrographic regimes surrounding the ice sheet (Fig. 1d), with a similar cutoffs beyond 1500 m bottom depth and geographical distance from the ice sheet to select coastal ocean cells near the ice sheet. For each sub-region, volume-averaged temperatures below 200 m depth are computed, providing a scalar near-shore sub-surface temperature metric. For Antarctica, the full depth range down to 1500 m is included, while for Greenland, the profiles are truncated below 500 m depth to account for shallow continental shelf depths and bottom sills that typically prevent inflows from greater depths toward the marine terminating glaciers in Greenland fjords.

Regional volume-averaged temperatures are also computed from available observed ocean climatologies, using the same algorithm as for the model output. For Greenland, observational data are taken directly from the annually averaged statistical fields of the 2013 World Ocean Atlas (Locarnini and Seidov, 2013). For Antarctica, a refined climatology of coastal water masses was constructed by combining the 2018 WOA data (WOA, Locarnini et al., 2019) with statistical fields from the EN4 ocean climatology (Good et al., 2013) and publicly available temperature profiles from Satellite Relay Data Logger–equipped seals (Roquet et al., 2018), with further details provided in Jourdain et al. (in prep). In both cases, ocean measurements close to the ice sheets are so sparse that all observations are included in the computation of the regional averages, regardless of their acquisition date.



### 2.2.3 Aggregating historical metrics

In order to aggregate different metrics of varying nature and magnitude, each of the historical metrics $\chi$ described above, is normalized with regards to the 33-model multi-model median and interquartile range (IQR). For each model i:

$$\chi_{i,\,norm} = \frac{\chi_i - \mathrm{median}(\chi)}{\mathrm{IQR}(\chi)}. \tag{1}$$

We average the normalized metrics into three realms: atmosphere, surface ocean (for Antarctica), and sub-surface ocean. This decision was made to weaken the dependence of the final ranking on the number of variables used for each realm. Normalization of metrics prevents highly variable or large-amplitude metrics from being overly influential in the average (see Fig. A.1) while still penalizing extremes. The final aggregated score for each model is obtained by averaging atmosphere and

ocean for Greenland, and atmosphere, surface ocean, and sub-surface ocean for Antarctica. An alternative aggregating method, where all normalized metrics are weighted equally (12 for Antarctica, 7 for Greenland), is presented in Fig. A.2, and does not change our conclusions.

### 2.3 Projected 21C changes

### 2.3.1 Climate change metrics

For atmospheric and surface ocean variables, climate change metrics are calculated as the difference between the 2070-2100 mean (RCP8.5) and the 1980-2010 mean (historical) value of each variable, spatially-averaged over the entire Greenland and Antarctic atmospheric domains (Fig 1). For the sub-surface ocean, we define metrics as the change in volume-averaged regional temperature between the 1989-2009 and 2080-2100 periods. For Antarctica, we consider four metrics for the atmosphere (change in annual air temperature at 850 hPa, $\Delta$ta850[a], in annual precipitable water, $\delta$prw[a], and in position and strength

of the tropospheric westerly jet, $\Delta$Jpos and $\Delta$Jstr), two metrics for the surface ocean (change in winter sea ice concentration, $\Delta$sic[w], and in summer sea surface temperature, $\Delta$tos[s]), and six metrics for change in sub-surface ocean temperature, one for each of the sectors defined in Section 2.2.2. For Greenland, we define two metrics for the atmosphere (change in annual air temperature at 700 hPa $\Delta$ta700[a], and in annual precipitable water, $\delta$prw[a]) and four metrics for change in sub-surface ocean temperature, one for each ocean sector defined in Section 2.2.2.

### 2.3.2 Maximizing diversity of projected 21C changes

To maximize the diversity of future projections covered in a sub-selection of models of size $n$, we define the ensemble inter-model spread $E$ by combining the pairwise model differences across the climate change metrics defined in Section 2.3.1 (12 for Antarctica, 6 for Greenland). The spread of a 3-model ensemble is computed as the following:

$$E_{n=3} = \sum_{\chi} |\chi_{model\ 1} - \chi_{model\ 2}| + |\chi_{model\ 2} - \chi_{model\ 3}| + |\chi_{model\ 1} - \chi_{model\ 3}|, \tag{2}$$

with $\chi$ the climate change metrics defined in Section 2.3.1. The ensemble that maximizes $E$ for a given ensemble size $n$ ($n = 3$ for top 3, $n = 6$ for top 6) is the one qualified as 'most diverse' in its future projections.





## 3    Results for Antarctica

In this section, we focus on the model selection for the Antarctic ice sheet, which is based on historical ranking (Section 3.1) and projection diversity (Section 3.2). The selected models are presented in Section 3.3.

### 3.1    Historical bias ranking

Over the Antarctic domain, the total normalized historical metric ranges between -0.32 (model of highest fidelity, CanESM2) and 1.50 (model of lowest fidelity, BMU-ESM), with a median value of 0.13. Figure 2 shows the 33 climate models ranked by their historical metric, together with contributions of the sub-surface ocean, atmosphere and surface ocean to the total historical metric.

Models do not perform equally across the three realms. For example, GFDL-CM3 and EC-EARTH perform well in the atmosphere, with atmospheric metrics of -0.22 and -0.21 respectively, amongst the best models, but are ranked as low fidelity (with total bias scores of 0.46 and 0.54) due to their poor performance in ocean sub-surface and surface conditions. Conversely, IPSL-CM5B-LR performs well in the sub-surface ocean (metric of -0.20) but is penalized by its poor performance in the atmosphere (metric of 2.07) and surface ocean conditions (metric of 1.77).

Models also do not perform equally within each realm, indicating that biases originate due to regional processes for sub-surface ocean, or variable-specific biases for surface ocean and atmosphere. We provide the per-variable breakdown of the ocean sub-surface metric (Fig. 2b), and ocean surface and atmospheric metrics (Fig. 2c). Although this paper cannot address these differences in detail, we highlight a few notable sources of discrepancies between metrics. For example, the sub-surface heat in the Weddell Sea region is the largest single contributor to the ocean bias metric in several models (Fig. 2b), including EC-EARTH, MRI-CGM3 and BNU-ESM. The large ocean heat bias would warrant specific studies investigating the model representation of the ocean climatology in that region. Similarly, in the atmosphere, precipitable water is the largest single bias for models such as IPSL-CM5B-LR, INM-CM4, and MRI-CGCM3, and would warrant further investigation to improve model representation of the historical period.

Models that perform better than the median (historical metric < 0.13) have reasonable values for all three realms: the worst metric for each realm is lower than 50% of the IQR away from the ensemble median for that realm (Fig. 2a). This result gives confidence that these models have a good overall performance, rather than compensating biases across realms. Our averaging method was effective in penalizing models that have a low fidelity over an entire realm. For this reason, selecting the top 3 models in the top half of the 33 model ensures overall good performance of these models in both the ocean and atmosphere.

### 3.2    Projected changes

All 33 models considered in this study show an increase in air temperature over the Southern Ocean and Antarctic continent between the end of the 21st century and the end of 20th century climatologies (Fig. 3a), with a multimodel mean increase of 2.54 °C. Nevertheless, the ensemble shows a spread of transient climate sensitivity, with an atmospheric warming ranging from 1.3 °C (GFDL-ESM2G) to 3.6 °C (BNU-ESM), with a median of +2.5 °C. We highlight the AOGCMs selected in Section 3.3,





to illustrate the spread that they cover compared to the 33-model ensemble. Although the projected change in air temperature
is only one of the variables we use to diagnose projected atmospheric changes, it provides a good representation of projected
changes in the atmosphere. Indeed, the changes in annual air temperature are strongly correlated ($R^2 > 0.82$) to the projected
changes in seasonal air temperature, in annual and seasonal precipitable water, and strongly anti-correlated to changes in winter
sea ice extent ($R^2 = 0.70$). Projected changes in wind jet strength, as quantified in Bracegirdle et al. (2018), show a weaker
negative correlation with air temperature changes, although a decrease in jet strength is generally associated with a decrease in
annual sea ice extent ($R^2 = 0.46$), as noted in Bracegirdle et al. (2018).

Climate models also overwhelmingly project a 21st century increase in ocean temperatures around Antarctica. For example,
the 33 models project a warming of the Amundsen shelf (Fig. 3b), ranging from no significant warming (lowest warming, MRI-
CGCM3) to +1.10 °C (highest warming, IPSL-CM5B-LR), with a median value of +0.45 °C. In Fig. 3b, the models selected in
the top 3 and top 6 are highlighted in red and yellow respectively, to illustrate the spread that they sample over the Amundsen
region. Other regions show qualitatively similar range of projected changes, with the highest warming (as quantified by the
median value of the ensemble) occurring in the Dronning Maud Land, Amery, and Totten regions (DML median: +0.76 °C;
Amery median: +0.70 °C; Totten median: +0.59 °C). The lowest projected warming occurs in the Weddell and Ross regions
(Weddell median: +0.21 °C; Ross median: +0.30 °C). The Amundsen region, presented in Fig. 3b, is currently under scrutiny
due to ice-shelf thinning and accelerating ice discharge in the last decade (Kimura et al., 2017; Mouginot et al., 2014; Barletta
et al., 2018), but is only projected to warm moderately in the future according to the 33-model ensemble (Amundsen median:
+0.45 °C).

Unlike the atmospheric warming, which is a good proxy for other atmospheric changes, the projected ocean warming in the
Amundsen region is only weakly correlated ($R^2 \leq 0.016$) to other ocean regions. Some significant correlation can be found
for neighboring regions in East Antarctica, such as between the Dronning Maud Land and Amery regions ($R^2 = 0.71$) and
between the Amery and Totten regions ($R^2 = 0.48$), but is low across other regions ($R^2 \leq 0.25$). Projected changes in the
ocean are relatively independent across regions (detailed in Fig. B.1), which confirms the added value of quantifying regional
ocean metrics rather than metrics integrated over all Antarctic shelves.

### 3.3  Recommended ensemble

#### 3.3.1  Top 3

In the case of the Antarctic domain, the selection criteria described in Section 2 led to 6 suitable coupled models to choose from
(CanESM2, NorESM1-M, CSIRO-Mk3-6-0, CCSM4, MIROC-ESM-CHEM, MIROC-ESM), where availability of required
data from RCP2.6 projections is the strongest constraint. We then select the 3 models that maximize the ensemble diversity
$E_{n=3}$, as defined in Section 2.3.2. The selection is robust to removing one of the metrics at a time and to changing the weight
of the metrics in the calculation (Appendix C1).

The top 3 models selected are, in alphabetical order, CCSM4, MIROC-ESM-CHEM, and NorESM1-M. These 3 models
sample different projected changes in Antarctica under the RCP8.5 scenario (Fig. 4a). Overall, NorESM1-M shows a stronger





end-of-21st-century ocean warming than the ensemble median, but a low atmospheric warming compared to the model ensemble. Conversely, MIROC-ESM-CHEM features an ocean warming similar to that of the ensemble median, associated with strong atmospheric changes, about one IQR higher than the median. Finally, CCSM4 shows very distinct regional patterns
of ocean warming, with strong warming in the Weddell and Totten regions, and lower warming in the Ross and Dronning Maud Laud regions, relative to the ensemble median. The projected atmospheric changes in CCSM4 are on the high end of the ensemble, qualitatively similar to that of MIROC-ESM-CHEM.

### 3.3.2 Top 6

For the additional 3 models (targeted), CSIRO-Mk3-6-0 is chosen because of its good ranking (Fig. 2) and median projected
changes (Fig. 3,4b), and is preferred to ACCESS1.0, of similarly median projections under RCP8.5, because of the availability of the RCP2.6 scenario. Each of the metrics of future change lies close to the multi-model ensemble median (see Fig. 4b), meaning that approximately half of the 33 climate models predict higher changes than those of CSIRO-Mk3-6-0, and half predict lower changes.

The other two models selected are, in alphabetical order, HadGEM2-ES and IPSL-CM5A-MR. HadGEM2-ES brings diver-
sity to the 6-model ensemble because of its extreme end-of-21st-century warming in the ocean, particularly in the Ross Sea. This extreme regional warming, more than 2 times larger than the IQR from the median value, is ruled out of the top 3 because it is considered to be a less likely response than those produced by a high number of distinct climate models. Nevertheless, in an intercomparison effort such as ISMIP6, sampling high-end scenarios is essential to (i) examine the response of ice-sheet models which may have run-away effects, (2) include high risk (low probability, high cost) scenarios in terms of future sea level
rise. The atmospheric changes produced by HadGEM2-ES are higher than the median, but not outliers. Finally, IPSL-CM5A-MR features an ocean warming lower than the ensemble median in most ocean regions, and atmospheric changes higher than the median. It is the only model selected with systematically low warming in the ocean, and can be thought of as the converse to NorESM1-M. Robustness of the model selection is demonstrated in Appendix C2.

## 4 Results for Greenland

In this section, we describe the model selection for the forcing of the Greenland ice sheet. The methods include the model evaluation (included below) and ensemble selection (Section 4.2), mirroring the selection performed for the Antarctic ice sheet (Section 3).

### 4.1 Historical bias ranking

Coupled climate models do not perform equally over the sub-surface ocean and the atmosphere (Fig. 5a) around Greenland,
consistent with finding for Antarctica, shown in Section 3. Some models perform well in the atmosphere but are penalized by their poor ocean performance. For example, CMCC-CMS is the median of the ensemble and features one of the lowest biases in the atmosphere (-0.69) and one of the highest biases in the ocean (0.73). Conversely, others perform well in the ocean but





show high biases in the atmosphere (e.g. MRI-CGCM3). This unequal performance across the ocean and atmospheric variables supports the need to assess several components of coupled climate models together, rather than separately.

Investigating the source of biases in any given model is beyond the scope of this paper, which focuses on selecting 6 models suitable for the ISMIP6 simulations. Nevertheless, the ranking of the models can highlight significant biases. For example, the ocean bias in several models, most notably CMCC-CS, CMCC-CESM and IPSL-CM5B-LR, is dominated by a bias in ocean heat in the Arctic region. This large bias in temperature would warrant a specific study to improve model representation of that region. However, the observations in this region are scarce and we have a lower degree of confidence in the resulting ocean

climatology in that region than in more frequently and densely observed regions, as discussed in Section 5.

    The model ranking around Greenland highlights that the fidelity of coupled models is regionally dependent. The models of highest fidelity around Greenland do not necessarily perform well around Antarctica, and vice versa. For example, CanESM2 is the best-ranked model for Antarctica (see Section 3) but is ranked in the lower half of the ensemble around Greenland due in part to its ocean biases. Likewise, MIROC5 performs well on all metrics around Greenland, and has been extensively

used in the relevant literature (e.g., Fettweis et al., 2013; Tedesco and Fettweis, 2012), but has strong atmospheric biases over Antarctica. Climate models are not expected to perform equally in all regions, nevertheless, it is important for the scientific community to keep those regional variations in mind, especially if using existing studies performed over a different region. This unequal performance across the Greenland and Antarctic regions also supports our decision to perform model ranking and selection independently for the two ice sheets.

Finally, the models that perform better than the median have ocean and atmosphere biases that lie lower than 0.5 IQR away from the median. Although biases in individual (regional) variables may be higher than that, this result confirms that the best ranked models have a good performance in both the sub-surface ocean and the atmosphere, and gives us confidence that the top half of the ensemble are suitable candidates for the Greenland model selection.

### 4.2   Future projection diversity

All 33 AOGCMs project atmospheric warming over Greenland by the end of the 21st century. Projections range from +1.95 °C (lowest warming, FIO-ESM) to +5.95 °C (highest warming, MIROC-ESM-CHEM) with a median warming of +4.09 °C (Fig. 6a). Models that made our final selection, highlighted in red (top 3) and yellow (top 6), sample a range of future warming. Similar to results presented for Antarctica (Section 3), the changes in annual air temperature over Greenland are a good proxy for most other atmospheric changes. Increase in $700 hPa$ air temperature is associated with an an increase in precipitable water

($R^2 = 0.96$), an increase in ocean surface temperature ($R^2 = 0.60$), and a decrease in summer sea ice cover ($R^2 = 0.29$).

    Most models also project an increase in ocean temperature on the shelf surrounding Greenland. Baffin Bay, for example, is projected to warm by +0.48 °C by the end of the 21st century, with models projecting between +0.07 °C (lowest warming, BCC-CSM1-1) and +1.70 °C (highest warming, CanESM2). The models selected in Section 4.3, highlighted in Fig. 6, cover a range of projected warming in Baffin Bay. Two other regions show similar projected changes (Arctic median: +0.48°C;

Subpolar Gyre (SPG) median: +0.49°C). The highest projected warming occurs in the Greenland-Iceland-Norwegian region (GIN), with a median warming of +0.76°C.





Projected changes across the ocean regions are correlated between the Arctic Ocean and GIN regions ($R^2 = 0.58$), and between the SPG and GIN regions ($R^2 = 0.31$). Other regions are only weakly correlated with each other (detailed in Fig. B.2), and ocean changes show no significant correlation with the projected atmospheric changes ($R^2 < 0.06$).

## 4.3 Recommended ensemble

In the case of Greenland, the availability of sub-daily wind outputs is a strong constraint for the model selection. This was a determining factor because existing studies over Greenland show that the regional atmospheric model MAR outperforms climate models in representing realistic surface mass balance (e.g., Noël et al., 2018; Fettweis et al., 2013).

### 4.3.1 Top 3

When applying the selection criteria described in Section 2 and removing CNRM-CM5 due to unavailable data, 6 models remain for the top 3 selection (MIROC5, IPSL-CM5A-MR, NorESM1-M, ACCESS1-0, ACCESS1-3, HadGEM2-ES). In this case, MIROC5 was pre-selected, as it features changes similar to that of the ensemble median, meaning half of the models project stronger changes than those of MIROC5, and half project weaker changes. Two additional models are selected, maximizing ensemble diversity of three models ([MIROC5, model 1, model 2]).

The top 3 models selected are, in alphabetical order, HadGEM2-ES, MIROC5 and NorESM1-M. These 3 models show different patterns of projected changes by the end of the 21st century (Fig. 7a). As described above, MIROC5 is chosen as a good representation of the overall ensemble. HadGEM2-ES features high atmospheric changes, including increases in air temperature and precipitable water, of magnitude stronger than the ensemble median. The projected changes in ocean heat are more regionally dependent, with warming higher in the Arctic and GIN (north-east), and lower in Baffin Bay and SPG (south-west) relative to the ensemble median. Conversely, NorESM1-M features a warming in the atmosphere on the low-end of the 33-model ensemble projections. The ocean warming is also regionally dependent, with NorESM1-M featuring low warming in GIN, the Arctic, and the SPG regions and a strong warming in the Baffin Bay region.

### 4.3.2 Top 6

For the top 6 selection, 5 models (IPSL-CM5A-MR, CSIRO-Mk3-6-0, CCSM4, ACCESS1-0, ACCESS1-3) are available to complement the already selected top 3.

The selected models are, in alphabetical order, ACCESS1-3, CSIRO-Mk3-6-0, and IPSL-CM5A-MR. CSIRO-Mk3-6-0 projects a low atmospheric warming, far below the median value, alongside an extreme warming in the south-west ocean regions ($\Delta T$ Baffin Bay $> 2$; $\Delta T$ SPG $= 0.94$). ACCESS1.3 adds diversity to the ensemble as it shows strong warming in Baffin Bay and the Arctic Ocean, but low warming in the subpolar gyre region (SPG). Its atmospheric warming is close to the median. Finally, IPSL-CM5A-MR project strong warming in the Greenland-Iceland-Norwegian seas (GIN), while other ocean regions and atmospheric variables are closer to the median.





## 5  Discussion

In this study, we evaluated the performance of 33 CMIP5 AOGCMs relative to reanalyses and gridded observational datasets covering the atmosphere, sea surface, and sub-surface ocean around the Greenland and Antarctic ice sheets. We also assessed

21st century changes in key oceanic and atmospheric variables. Time constraints for ISMIP6 simulations drove several decisions relating to the scope of this analysis, including: the use of the CMIP5 (rather than the now-partially-available CMIP6) ensemble; the use of AOGCMs that had already been processed and regridded for both the ocean and atmosphere; and the use of available observational products with limitations and biases, particularly in the ocean sub-surface. However, this assessment of near-ice sheet present-day and future climate remains the most comprehensive performed to date.

Many subjective choices were made in the model selection process. We have attempted to document these choices, and note that the relative insensitivity of results to alternate choices (e.g., Fig. A2, Appendix C) provides some confidence that our rankings are robust for the CMIP5 ensemble. However, because the rankings will not be applicable to future model ensembles (e.g. CMIP6), our discussion focuses on key elements of our methodology that could be further developed. Implications are discussed with respect to results from the full 33-member ensemble to extend the relevance to other exercises where the small

ensemble required for ISMIP6 may not apply.

Model selection was made largely based on their representation of the present-day local climate, with the implicit assumption that biases relative to observations reflect a poor representation of processes of relevance to future warming. It is difficult to determine whether performance relative to this set of present-day regional metrics is: 1) a sufficient means to evaluate AOGCMs and 2) relevant to the rate of 21st century near-ice-sheet warming. Krinner and Flanner (2018) shows that model

biases are stationary under future climate change within the CMIP5 dataset, providing justification for using less biased models for climate change studies. However, over the long timescales that ISMIP6 seeks to assess, different processes and/or biases (global and/or non-local ocean warming rates, e.g. stratospheric ozone recovery) may be equally important; i.e., even if a model closely matches historical conditions it may be missing a key process important for projections.

Support for the relevance of these metrics might be derived from a clear relationship between the modern state and projec-

tions of change across models (so-called "emergent constraints"). Bracegirdle et al. (2015) and Agosta et al. (2015) found that 21st century changes in Antarctic air temperature and precipitation rate (and, perhaps surprisingly, jet strength (Bracegirdle et al., 2018)) were correlated to sea ice area bias across models. In this analysis, we found no significant correlation between historical biases and climate changes over Antarctica (or Greenland). A plausible explanation is our use of an 850 hPa (rather than surface) temperature metric and our circum-Antarctic study region. However, this result may also indicate a sensitivity

to the specific models included in the ensemble: we find that the magnitude and significance of inter-model correlations are sensitive to whether all or a set of the best-performing models is assessed. Shared code and parameterizations across models may also underlie some of the modest correlations evident in our analysis.

It is difficult to determine whether the historical metrics chosen in this analysis are comprehensive (e.g. account for all relevant processes) and/or independent. With respect to independence, we eliminated metrics which respresent the same phys-

ical processes and are strongly correlated (e.g., the precipitation and air temperature variables in Bracegirdle et al. (2015) are





strongly correlated to those in Agosta et al. (2015) and were not included in this study). Assessing comprehensiveness is more difficult. For example, the choice of metrics is constrained by the availability of observations. In particular, oceanographic measurements in the vicinity of ice sheets are very sparse and feature sharp horizontal gradients in water masses (e.g., Thompson et al., 2018). As a result, we chose to calculate volume- and time- mean quantities over subjectively defined regions in order to

maximize the number of observations included. It is unclear which ocean region is most "important" in terms of future mass balance. The optimal number of regions, based on their relevance to future ice sheet change and their independence, remains to be determined. These choices should be expected to influence both evaluations of performance and warming. In contrast, observations for the atmosphere and surface ocean have better spatiotemporal coverage. Correspondingly, the metrics chosen were continental-scale and seasonally resolved. However, our continental-scale evaluation may obscure regional variability.

Future work should more formally assess the number and relative weighting of regional metrics in the atmosphere and ocean. Similar concerns apply to the metrics of future warming, and their relevance to ice sheet mass balance. We note that our analysis does not address the rate of warming, which differs widely across models. In the ocean, the rate and timing of warming may have dramatic effects on 21st century ice sheet evolution (Hellmer et al., 2012; Timmermann and Goeller, 2017).

We have noted the unequal performance of coupled climate models over different realms, which we suggest highlights the
importance of assessing model fidelity over a range of metrics combining the sub-surface ocean, surface ocean and atmosphere conditions. It also explains why the present ranking of models differs from existing intercomparison studies specifically focused on the atmosphere (Agosta et al., 2015, e.g.,) or the ocean (e.g., Sallée et al., 2013; Meijers et al., 2012; Russell et al., 2018). For example, the metrics used in Agosta et al. (2015) led to EC-EARTH and CanESM2 being ranked closely (8 and 9 out of 41 models), implying similar performance. However, by including the sub-surface ocean metrics, our results point to CanESM2
as the model with the best fidelity overall, while EC-EARTH is in the lower half of the 33-model ensemble due to its poor performance in the ocean (other examples of differences in rankings across realms can be found by examining Fig. 2 or Fig. 5). As Agosta et al. (2015) focuses purely on the model performance for ice-sheet surface mass balance, its results differ from this paper evaluating both the ocean and atmospheric metrics for the sake of providing the atmosphere-driven surface mass balance and the ocean-driven melt from the same coupled model as boundary conditions to ice-sheet models. This underscores the
importance of considering the original aim of an intercomparison, including the variables and the regions considered, before interpreting or applying ranking derived from the analysis.

Antarctica and Greenland were treated independently, supported by the different performance across the ensemble. A different set of models was selected for Greenland and Antarctica, suggesting model performance varies in polar regions of different hemispheres. However, with respect to future warming, it is reasonable to expect some degree of inter-hemispheric correlation
in warming (e.g. due to a high AOGCM climate sensitivity). It is unclear how this inter-ice sheet independence assumption could influence sea level projections, as it depends upon the response of SMB and changes in ice flux in the different ice sheets.

Using aggregated measures of present-day performance and future climate changes, we selected 6 AOGCMs as adequate and representative of future near-ice sheet warming pathways. This ensemble size was judged to be feasible for ISMIP6, given computational limitations and the goal to sample different sources of uncertainty (e.g. model, RCP, parameterizations, parameters,
etc). However, given the many degrees of freedom across the evaluation metrics, it is difficult to select a fully representative
sample. Some limitations of the sample size are apparent, notably the non-uniform distribution across parameters (e.g. no low ocean warming sampled). Furthermore, the models selected are not structurally independent. For example, HadGEM2-ES and ACCESS-1.3 share a common Hadley Center atmospheric model, while NorESM1 and CCSM4 share the NCAR Community Atmospheric Model. Such interdependence may limit the diversity of forcing applied to ISMIP6 models. We do note that even
if ISMIP6 had the ability to evaluate all available CMIP5 AOGCMS, issues with statistical sampling and diversity of CMIP models, code similarities/independence, and quality would persist (Knutti et al., 2013; Sanderson et al., 2015a, b). Future model evaluation studies may invert the process used here: i.e., objectively assess the appropriate number of models to achieve sufficient diversity in forcing.

Finally, we emphasize that evaluation is only a first step to a better process-based understanding of the differences between
models. It is critical to assess the processes that make models (or model families) perform better or warm at a different rate. We invite modeling groups or researchers interested to examine these to trace back the source of the bias in individual models or across the larger ensemble.

## 6   Conclusions

As a result of the evaluation and selection process described in this paper, six AOGCMS have been selected for ISMIP6 Antarc-
tic future projection runs, and six AOGCMS have been selected for ISMIP6 Greenland future projection runs. To complement the quantitative comparison described in the results section, a qualitative description of their projected warming is shown in Table 2, and Table 3. It is expected that the range of near-ice sheet climate changes simulated by these AOGCMs will result in diverse projections of ice sheet mass balance change when used to force ISMIP6 simulations. Readers interested in the translation of AOGCM output to forcing for ISMIP6 simulations are encouraged to Slater et al. (2019), Jourdain et al. (in prep)
and Nowicki et al. (in prep).

The AOGCMs selected for ISMIP6 Greenland projection runs, and their qualitative projected warming, are summarized in Table 3

**Appendix A:  Robustness of historical ranking**

**Appendix B:  Projected 21C ocean warming**

**Appendix C:  Robustness of model selection**

This appendix describes robustness of the model selection to modifications of the choice and weight of metrics. We repeat the model selection for the top 3 and top 6 models for Antarctica (Sec. 3.3) and Greenland (Sec. 4.3) under removal of one of the metrics at a time and under a change of the weighting. Overall, the model selection is robust to the described modifications.





## C1 Robustness of Antarctic model selection top 3

Table C1 lists the selected model combinations with absolute and relative frequency of occurrence for the Antarctic top 3 selection. The final model combination (NorESM1-M, MIROC-ESM-CHEM, CCSM4) occurs in 9 of 12 cases. One additional model (CanESM2) is selected in 25 % of the cases. Table C2 lists the absolute and relative occurrence of each individual model in the combinations in Table C1.

## C2 Robustness of Antarctic model selection top 6

Table C3 lists the selected model combinations with absolute and relative frequency of occurrence for the Antarctic top 6 selection. The final model combination (NorESM1-M, MIROC-ESM-CHEM, CCSM4, CSIRO-Mk3-6-0, HadGEM2-ES, IPSL-CM5A-MR) occurs in 12 of 14 cases. Table C4 lists the absolute and relative occurrence of each individual model in the combinations given in Table C3. When equal weighting of the 14 metrics is applied, giving more emphasis on the surface ocean, HadGEM2-ES is still selected in 4 of 14 cases, but replaced by MPI-ESM-MR in the majority of cases (9 of 14).

## C3 Robustness of Greenland model selection top 3

Table C5 lists the selected model combinations with absolute and relative frequency of occurrence for the Greenland top 3 selection. The final model combination (MIROC5, NorESM1-M, HadGEM2-ES) was selected in all cases. Table C6 lists the absolute and relative occurrence of each individual model in the combinations given in Table C5.

The same results were obtained when metrics for the surface ocean ($\Delta$tos [a], $\Delta$sic [s], $\Delta$sic [w]) were added to the other 450 metrics($\Delta$700hPa [a], $\delta$prw [a] , $\Delta$T SPG, $\Delta$T BB, $\Delta$T AO, $\Delta$T GIN).

## C4 Robustness of Greenland model selection top 6

Table C7 lists the selected model combinations with absolute and relative frequency of occurrence for the Greenland top 6 selection. The final model combination (MIROC5, IPSL-CM5A-MR, CSIRO-Mk3-6-0, NorESM1-M, HadGEM2-ES, ACCESS1-3) occurs in 7 of 9 cases, with CCSM4 replacing ACCESS1-3 in the remaining 2 cases. Table C8 lists the absolute and relative 455 occurrence of each individual model in the combinations given in Table C7.

Similar results were obtained whether metrics for the surface ocean ($\Delta$tos [a], $\Delta$sic [s], $\Delta$sic [w]) were included or not.

*Author contributions.* AB, CA, CML, NJ, TH, HG, HS, FS and SN designed the study and the evaluation methodology. AB, CA and HG performed analysis on data provided by CA, CML, NJ, TH and TJB. AB prepared the manuscript with contributions from all co-authors.

*Competing interests.* The authors declare that they have no conflict of interest.



*Data availability.* The supporting data is available at 10.5281/zenodo.3367347.

*Acknowledgements.* AB was supported by the U.S. Department of Energy (DOE) Office of Science Regional and Global Model Analysis (RGMA) component of the Earth and Environmental System Modeling (EESM) program (HiLAT-RASM project), and the DOE Office of Science (Biological and Environmental Research), Early Career Research program. CA was supported by the Agence Nationale de la Recherche Scientifique, project ANR-15-CE01-0015 (AC-AHC2), and by the Fondation Albert 2 de Monaco under the project Antarctic-

Snow (2018–2020). CML acknowledges financial support from NSF grants 1513396 and 1744792. TH acknowledges financial support from Norwegian Research Council projects 231549 and 280727. NJ's contribution was partly funded by the French National Research Agency (ANR) through the TROIS-AS (ANR-15-CE01-0005-01). HG has received funding from the program of the Netherlands Earth System Science Centre (NESSC), financially supported by the Dutch Ministry of Education, Culture and Science (OCW) under Grant nr. 024.002.001. HS was supported by grants from the NASA Cryospheric Science, Sea Level Change Team, and Modeling Analysis and

Prediction Program. FS acknowledges financial support from NSF grants 1756272 and 1916566. TJB acknowledges support from NERC grant NE/N018486/1 and the AntClim21 Scientific Research Programme of the Scientific Committee on Antarctic Research. All members of the ISMIP6 collaboration are thanked for discussions and feedback, with particular thanks to Donald Slater, and Denis Felikson.



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





**Table 1.** ERA-Interim reanalysis and CMIP5 models used in this study.

| Name | Modelling group | Atm. grid spacing | 6-hourly available | rcp26 available | rcp85 available |
|---|---|---|---|---|---|
| ERA-Interim | ECMWF | 0.7° | x | | |
| ACCESS1-0 | CSIRO-BOM | 1.25° | x | | x |
| ACCESS1-3 | CSIRO-BOM | 1.25° | x | | x |
| BCC-CSM1-1 | BCC | 2.8° | | x | x |
| BNU-ESM | GCESS | 2.8° | | x | x |
| CanESM2 | CCCma | 2.8° | x | x | x |
| CCSM4 | NSF-DOE-NCAR | 1.25° | x | x | x |
| CESM1-BGC | NSF-DOE-NCAR | 1.25° | | | x |
| CESM1-CAM5 | NSF-DOE-NCAR | 1.25° | | x | x |
| CMCC-CESM | CMCC | 3.75° | | x | x |
| CMCC-CM | CMCC | 0.75° | | | x |
| CMCC-CMS | CMCC | 1.8° | | | x |
| CNRM-CM5 | CNRM-CERFACS | 1.4° | x | x | x |
| CSIRO-Mk3-6-0 | CSIRO-QCCCE | 1.9° | x | x | x |
| EC-EARTH | EC-EARTH | 1.125° | | x | x |
| FGOALS-g2 | LASG-IAP | 2.8° | | x | x |
| FIO-ESM | FIO | 2.875° | | x | x |
| GFDL-CM3 | NOAA GFDL | 1.8° | x | x | x |
| GFDL-ESM2G | NOAA GFDL | 2.0° | x | x | x |
| GFDL-ESM2M | NOAA GFDL | 2.0° | x | x | x |
| HadGEM2-CC | MOHC | 1.25° | | | |
| HadGEM2-ES | MOHC | 1.25° | x | x | x |
| INM-CM4 | INM | 1.5° | | | x |
| IPSL-CM5A-LR | IPSL | 1.9° | x | x | x |
| IPSL-CM5A-MR | IPSL | 1.3° | x | x | x |
| IPSL-CM5B-LR | IPSL | 1.3° | x | | x |
| MIROC-ESM | MIROC | 2.8° | x | x | x |
| MIROC-ESM-CHEM | MIROC | 2.8° | x | x | x |
| MIROC5 | MIROC | 1.4° | x | x | x |
| MPI-ESM-LR | MPI-M | 1.9° | | x | x |
| MPI-ESM-MR | MPI-M | 1.8° | | x | x |
| MRI-CGCM3 | MRI | 1.1° | x | x | x |
| NorESM1-M | NCC | 1.9° | x | x | x |
| NorESM1-ME | NCC | 1.9° | | x | x |



**Figure 2.** (a) Ranking of models according to total bias (black) over the Antarctic domain, with a break-down of the ocean (blue), atmosphere (orange) and surface (yellow) contributions. (b) Break-down of model performance in the ocean over the Antarctic domain. (c) Break-down of model performance in the atmosphere (orange) and ocean surface (yellow) over the Antarctic domain. Models are ranked according to total bias, and markers (*,†) identify models selected in the top3 and top6 ensembles respectively.





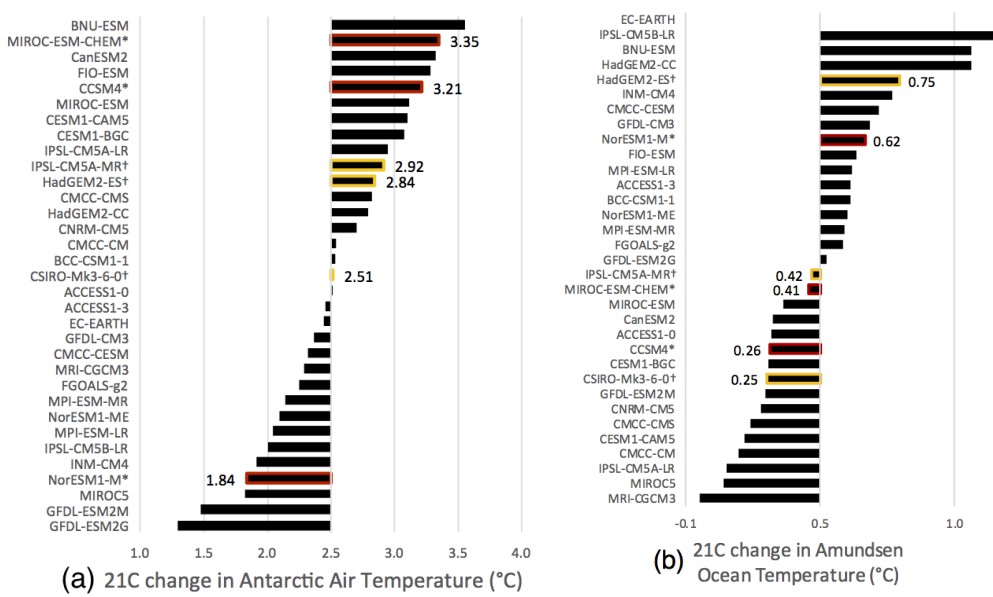

**Figure 3.** Projected RCP8.5 warming for each CMIP5 model in the Antarctic region. (a) Change in 850 hPa air temperature over the Southern ocean between 1980-2000 and 2080-2100. (b) Change in ocean temperature in the Amundsen region between 1980-2000 and 2080-2100. Models selected in the top3 (top6) ensemble are highlighted in red (yellow).

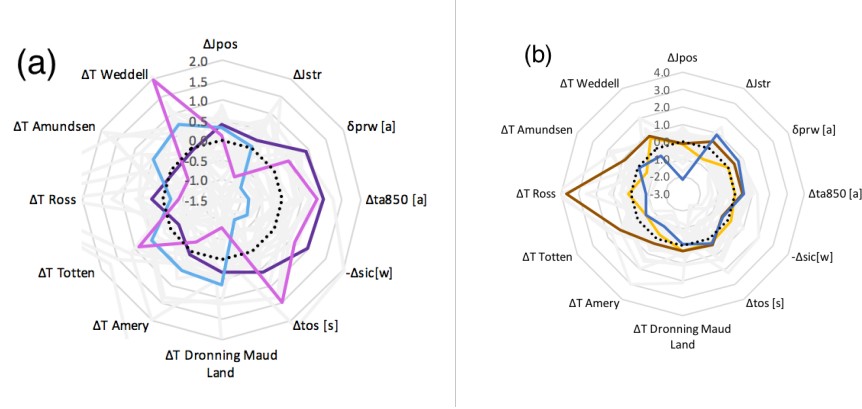

**Figure 4.** Normalized projected 21C changes for Antarctica (with model ensemble in gray and the median in black). (a) Top 3: CCSM4 (pink), MIROC-ESM-CHEM (purple), and NorESM1-M (light blue); (b) Top 4-6: CSIRO-Mk3-6-0 (yellow), HadGEM2-ES (brown) and IPSL-CM5A-M (dark blue).



**Figure 5.** (a) Ranking of models according to total bias (black) over the Greenland domain, including the ocean (blue) and atmosphere (orange) contributions. (b) Break-down of model performance in the ocean over the Greenland domain. (c) Break-down of model performance in the atmosphere over the Greenland domain. Models are ranked according to total bias, and markers (*,†) identify models selected in the top3 and top6 ensembles respectively.



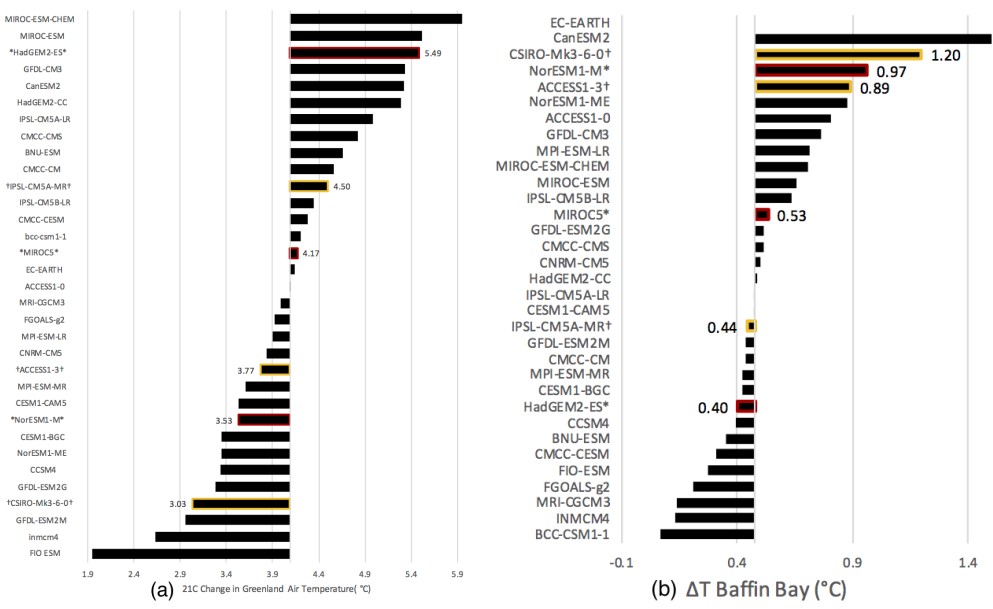

**Figure 6.** Projected RCP8.5 warming for each CMIP5 model over Greenland. (a) Change in 700 hPa air temperature over the Southern ocean between 1980-2000 and 2080-2100. (b) Change in ocean temperature in the Baffin Bay region between 1980-2000 and 2080-2100. Models selected in the top3 (top6) ensemble are highlighted in red (yellow).

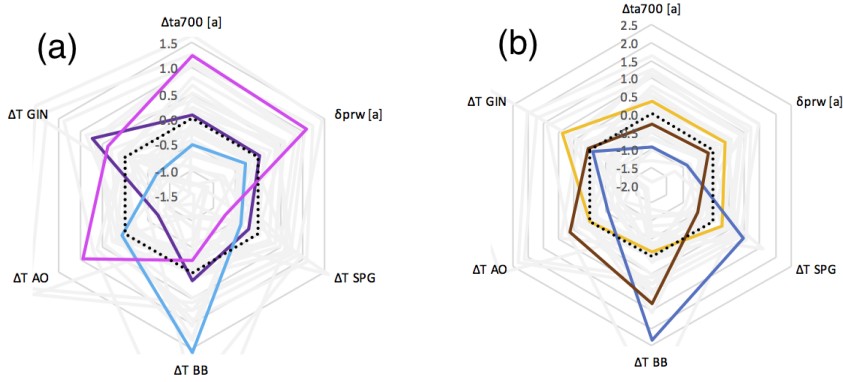

**Figure 7.** Normalized projected 21C changes for Greenland (with model ensemble in gray and the median in black). (a) Top 3: HadGEM2-ES (pink), MIROC5 (purple) and NorESM1-M (light blue); (b) Top 4-6: ACCESS1-3 (brown), CSIRO-Mk3-6-0 (dark blue), and IPSL-CM5A-MR (yellow). Ocean warming is calculated over 4 sectors (BB = Baffin Bay; AO = Arctic Ocean; GIN = Greeland-Iceland-Norwegian Seas; SPG = Subpolar Gyre).





**Table 2.** Selected AOGCMs for Antarctica and their qualitative projected warming

.

| Model | Ocean | Atmosphere | Comments |
|---|---|---|---|
| CCSM4 | median | high | Strong regional ocean differences |
| MIROC-ESM-CHEM | median | high | |
| NorESM1-M | mid-to-high | low | |
| CSIRO-Mk3-6-0 | median | median | |
| HadGEM2-ES | high | median | Extreme warming in the Ross Sea |
| IPSL-CM5A-M | low | high | |

**Table 3.** Selected AOGCMs for Greenland and their qualitative projected warming

.

| Model | Ocean | Atmosphere | Comments |
|---|---|---|---|
| HadGEM2-ES | high | low | Strong warming in Baffin Bay |
| MIROC5 | median | median | |
| NorESM1-M | median | high | Strong warming in the Arctic Ocean |
| ACCESS1.3 | mid-to-high | median | Strong warming in Baffin Bay |
| CSIRO-Mk3-6-0 | mid-to-high | low | Extreme warming in Baffin Bay |
| IPSL-CM5A-M | low | high | Strong warming in the GIN Seas |

**Table C1.** Top 3 selected model combinations for Antarctica with absolute and relative frequency of occurrence in the robustness test

.

| Model combination | Count | Occurrence |
|---|---|---|
| NorESM1-M, MIROC-ESM-CHEM, CCSM4 | 9 | 0.75 |
| CanESM2, NorESM1-M, CCSM4 | 3 | 0.25 |



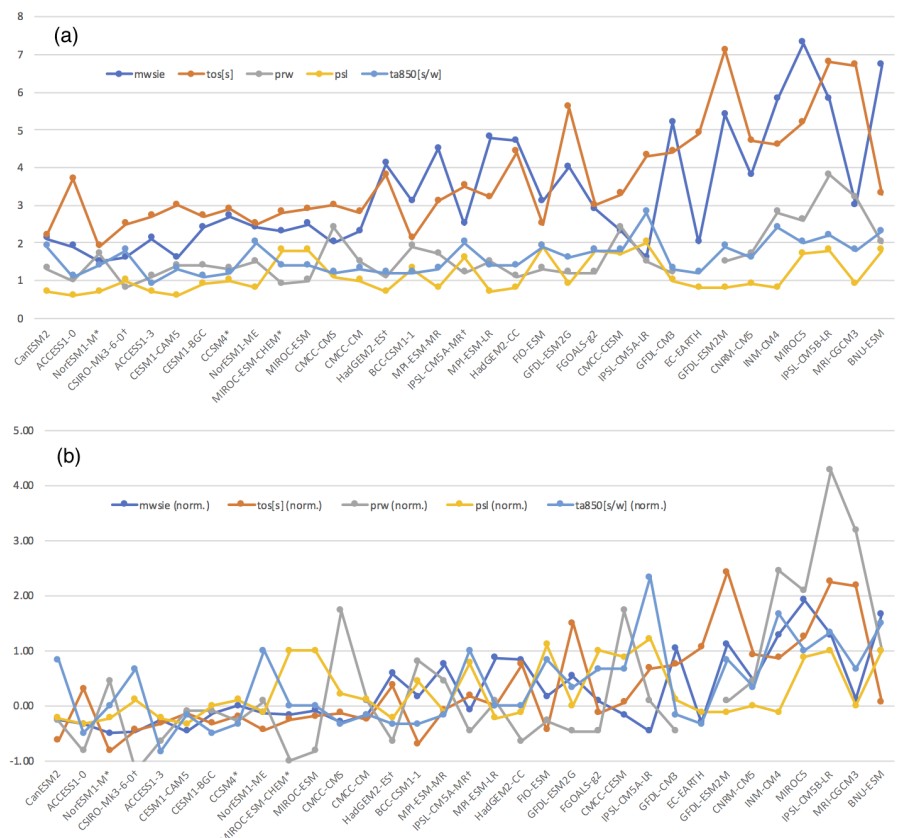

**Figure A.1.** Normalization of variables: (a) Historical atmospheric biases for the Antarctic domain; (b) Normalized historical atmospheric biases for the Antarctic domain. The non-normalized variables have different mean values, and different variability. The normalization removes the offset and rescales the variability, so that variables of different nature, magnitude, and variability can be combined in one atmospheric bias metric.

**Table C2.** Top 3 selected models for Antarctica with absolute and relative occurrence of each individual model in the combinations (Tab. C1)

.

| Models | Count | Occurrence |
|---|---|---|
| NorESM1-M | 12 | 1.00 |
| CCSM4 | 12 | 1.00 |
| MIROC-ESM-CHEM | 9 | 0.75 |
| CanESM2 | 3 | 0.25 |





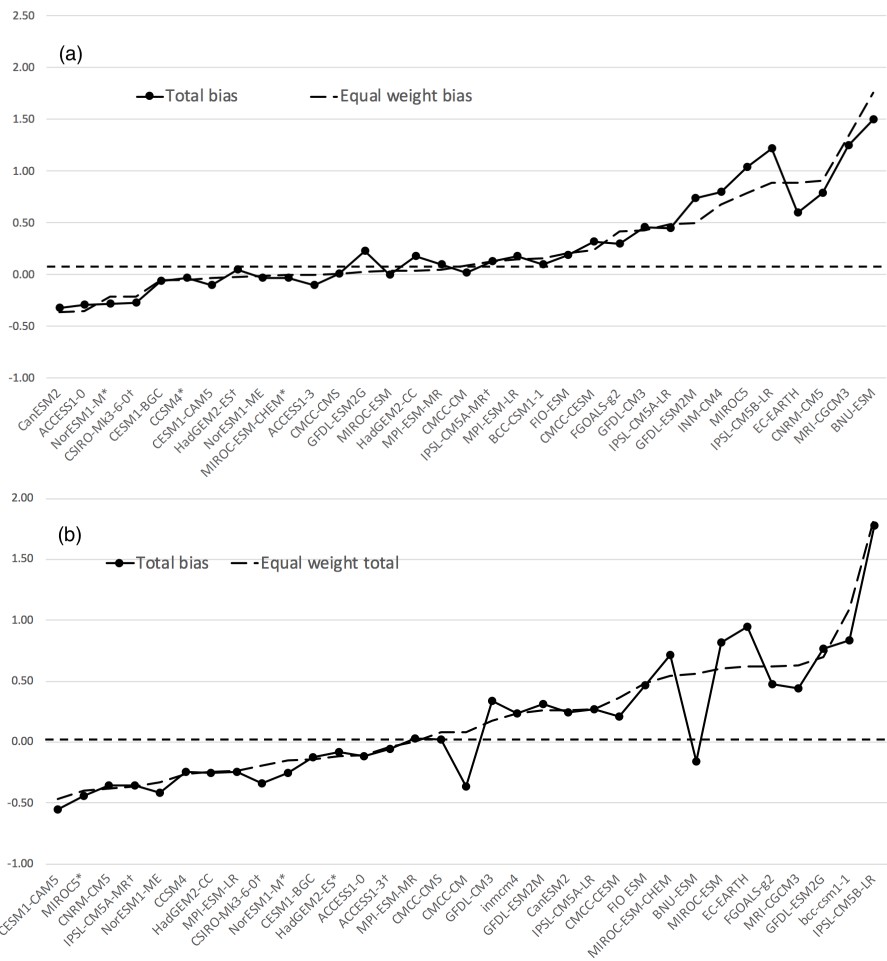

**Figure A.2.** Alternate ranking of AOGCMs according to an equal-weight total bias (dashed black) compared to the realms-averaged total bias (black) over (a) the Antarctic domain, (b) the Greenland domain.

**Table C3.** Top 6 selected model combinations for Antarctica with absolute and relative frequency of occurrence in the robustness test

| Model combination | Count | Occurrence |
|---|---|---|
| NorESM1-M, MIROC-ESM-CHEM, CCSM4, CSIRO-Mk3-6-0, HadGEM2-ES, IPSL-CM5A-MR | 12 | 0.86 |
| NorESM1-M, MIROC-ESM-CHEM, CCSM4, CSIRO-Mk3-6-0, BCC-CSM1-1, IPSL-CM5A-MR | 1 | 0.07 |
| NorESM1-M, MIROC-ESM-CHEM, CCSM4, CSIRO-Mk3-6-0,HadGEM2-ES, BCC-CSM1-1 | 1 | 0.07 |



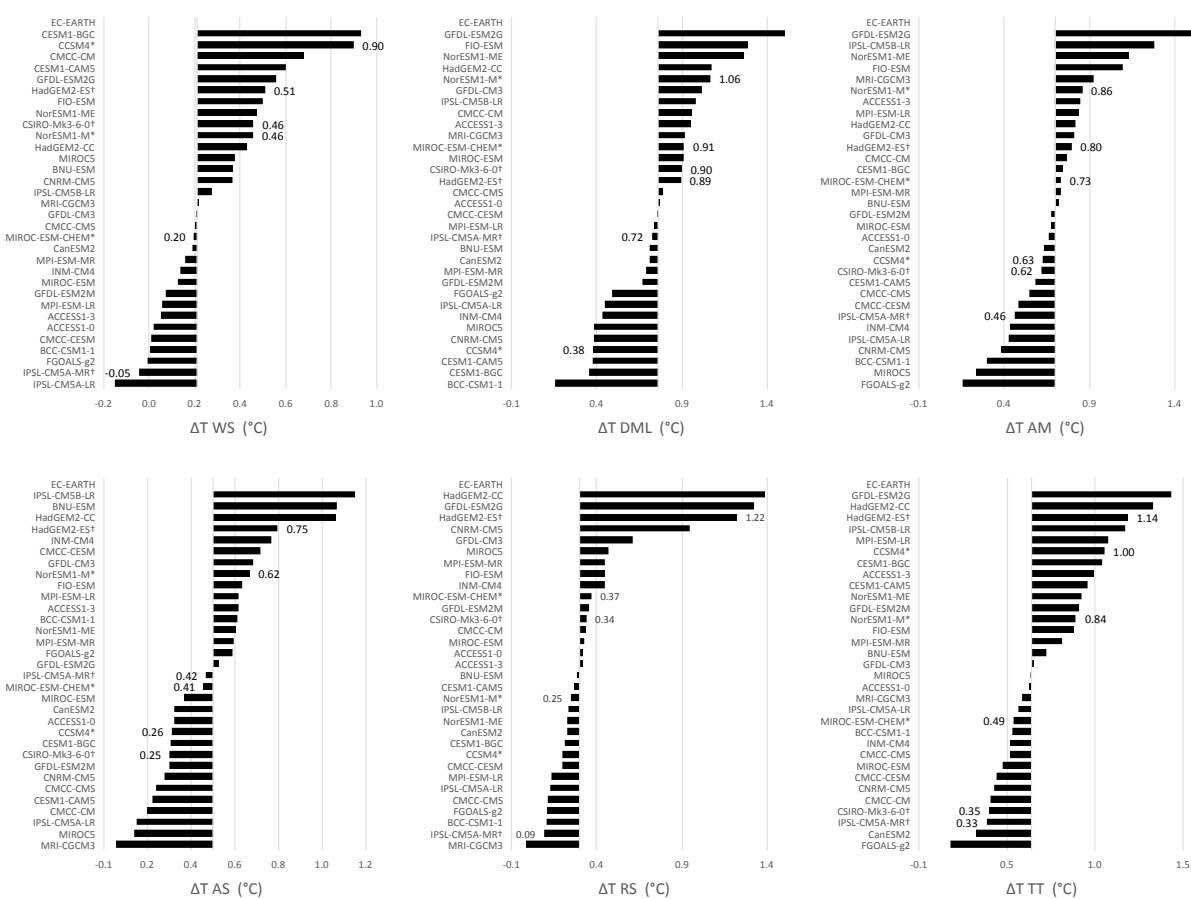

**Figure B.1.** Projected RCP8.5 warming for each CMIP5 model between 1980-2000 and 2080-2100 in the 6 Antarctic shelf regions (WS = Weddell Sea; DML = Dronning Maud Land; AM = Amery; AS = Amundsen; RS = Ross; TT = Totten). Labels and markers (*,†) identify models selected in the top3 and top6 ensembles respectively.





**Figure B.2.** Projected RCP8.5 warming for each CMIP5 model between 1980-2000 and 2080-2100 in the 4 Greenland shelf regions (BB = Baffin Bay; AO = Arctic Ocean; GIN = Greeland-Iceland-Norwegian Seas; SPG = Subpolar Gyre). Labels and markers (*,†) identify models selected in the top3 and top6 ensembles respectively.



**Table C4.** Top 6 selected models for Antarctica with absolute and relative occurrence of each individual model in the combinations (Tab. C3)

| Models | Count | Occurrence |
|---|---|---|
| NorESM1-M | 14 | 1.00 |
| MIROC-ESM-CHEM | 14 | 1.00 |
| CCSM4 | 14 | 1.00 |
| CSIRO-Mk3-6-0 | 14 | 1.00 |
| IPSL-CM5A-MR | 13 | 0.93 |
| HadGEM2-ES | 13 | 0.93 |
| BCC-CSM1-1 | 2 | 0.14 |

**Table C5.** Top 3 selected model combinations for Greenland with absolute and relative frequency of occurrence in the robustness test

| Model combination | Count | Occurrence |
|---|---|---|
| MIROC5, NorESM1-M, HadGEM2-ES | 9 | 1.00 |

**Table C6.** Top 3 selected models for Greenland with absolute and relative occurrence of each individual model in the combinations (Tab. C5)

| Models | Count | Occurrence |
|---|---|---|
| MIROC5 | 9 | 1.00 |
| NorESM1-M | 9 | 1.00 |
| HadGEM2-ES | 9 | 1.00 |

**Table C7.** Top 6 selected model combinations for Greenland with absolute and relative frequency of occurrence in the robustness test

| Model combination | Count | Occurrence |
|---|---|---|
| MIROC5, IPSL-CM5A-MR, CSIRO-Mk3-6-0, NorESM1-M, HadGEM2-ES, ACCESS1-3 | 7 | 0.78 |
| MIROC5, IPSL-CM5A-MR, CSIRO-Mk3-6-0, NorESM1-M, CCSM4, HadGEM2-ES | 2 | 0.22 |





**Table C8.** Top 6 selected models for Greenland with absolute and relative occurrence of each individual model in the combinations (Tab. C7)

| Models | Count | Occurrence |
| --- | --- | --- |
| MIROC5 | 9 | 1.00 |
| HadGEM2-ES | 9 | 1.00 |
| NorESM1-M | 9 | 1.00 |
| IPSL-CM5A-MR | 9 | 1.00 |
| CSIRO-Mk3-6-0 | 9 | 1.00 |
| ACCESS1-3 | 7 | 0.78 |
| CCSM4 | 2 | 0.22 |