# Peer review of "CMIP5 model selection for ISMIP6 ice sheet model forcing: Greenland and Antarctica"

_The Cryosphere, 2019_

## Referee Comment (RC1) · Anonymous Referee #1 · 25 Oct 2019

Summary This article summarizes the selection of atmosphere-ocean coupled climate models (AOGCMs) to use for forcing for the stand-alone ice sheet simulations as part of the CMIP6 ice sheet intercomparison project (ISMIP6). The manuscript summarizes the methods used to select the models and recommends 6 models each for use with Antarctica and Greenland ice sheets. The models used in the selection process are those from CMIP5 AOGCMs (CMIP6 were insufficiently available for testing at the time of this analysis). Three "core" models are chosen for both Antarctica and Greenland based on their fidelity to observations during the satellite record period (1979-2005). Three more models ("targeted") were selected for use based on representation of a range of future atmosphere-ocean conditions from both the RCP2.6 and RCP8.5 emissions scenarios. This submission documents the selection criteria and subsequent of

specific AOGCM selection for forcing of regional models for the ISMIP6. The work described is new and unique in that it uses both atmospheric and oceanic observations (rather than just atmospheric as in previous work) in the selection criteria. Antarctica and Greenland are treated separately, and with some overlapping and some unique variables as part of the evaluation of the AOGCMs.

The manuscript is well written, represents a significant scientific advance w.r.t. model selection for boundary conditions for ice sheet models. I recommend it be accepted for publication in The Cryosphere with minor revisions and technical corrections as follows:

Minor revisions: Much of the regional variability in Antarctica is related to the zonal asymmetry in the Southern Annular Mode (SAM; or likewise the depth, location, and seasonal migration of the Amundsen Sea Low, ASL). Some models do a better job than others at capturing this – which is different than the metrics of zonal jet location and strength. There are many atmospheric and oceanic metrics used to select the model criteria in this submission, although none directly measure whether or not the models capture asymmetric nature of the SAM (although the combination of oceanic and atmospheric metrics used may indeed capture it indirectly). A full analysis of this (whether or not models capture this asymmetry, not to mention how, exactly, to measure if the models do) is beyond the scope of this paper. I do feel, however, that some mention is worthwhile – do you believe your metrics indeed capture this even if indirectly? Or do you think some of the regional biases might be due to a particular model's lack of an ASL? A model's fidelity or lack thereof to ASL could help explain some of the regional discrepancies in projected changes as well. (e.g. M. Holland, L. Landrum, Y. Kostov and J. Marshall, 2016, Sensitivity of Antarctic sea ice to the Southern Annual Mode in coupled climate models, Clim. Dyn., DOI 10.1007/s00382-016-3424-9; J. T. M. Lenaerts, J. Fyke, B. Medley. The signature of ozone depletion in recent Antarctic precipitation change: a study with the Community Earth System Model, 2018. Geophys. Res. Lett., 45, 23, https://doi.org/10.1029/2018GL078608)

A couple sentences summarizing the figures/main point for each appendix would be

helpful (have one sentence for Appendix C, none for A, B).

Conclusions? Please finish!

Technical corrections (I can't figure out how to cut and paste Greek chi here so I write [chi]): Line 147: "historical metrics [chi] described above" but chi is not defined above. I believe [chi] in this case is the RMSE from the observations for each given variable – state this

Lines 315-322: Section 4.3 Top 3 (Greenland) Last sentence in first paragraph ("model 1, model2") sounds like a placemarker – eliminate or re-write

Lines 435-455 Check figure numbers. Mismatch between titles (in bold) and descriptions below (e.g. lines 439: "C2 Robustness of Antarctic. . .." Followed in line 440 by "Table C3 lists the . . .."

Figure 1. Regional oceanic boundaries (and some of the text over the map of the continent) for Antarctica are difficult to see (very difficult in the printed version – better on the screen) – recommend trying for different colors, or perhaps thicker outlines of the regions. The most difficult regional texts are "Weddell (WS)" followed by "Amery (AM) caption:"Greenland. . .inside the usual boundaries of MAR simulations" define MAR?

Figures 2, 3, 5, 6, A.1, A.2, B.1, B.2 The symbols denoting models that were in top3 and top6 ensembles are very difficult to see (and not stated in captions for A.1, A.2). Figures 3 and 6 highlight w different colors so perhaps not as important in these, however in the other figures these symbols need to be easier to spot – with color, or bold, or?

Table C2. Rewrite caption. . .says "three top models" and give statistics for four models (which are the four that give the two top-three combos). . .

---

## Referee Comment (RC2) · Anonymous Referee #2 · 27 Oct 2019

**Review of *"CMIP5 model selection for ISMIP6 ice sheet model forcing: Greenland and Antarctica"* submitted to TC by A. Barthel et al.**

With the aim of selecting a set of global climate models that represent best the current and projected climate of the Greenland (GrIS) and Antarctic ice sheets (AIS) to force the ice sheet models of ISMIP6, the authors evaluate, compare and rank 33 CMIP5 AOGCMs using observational data (present-day) combined with various atmospheric and oceanic metrics (scenarios). As a result, an ensemble of six AOGCMs (i.e. three core and three targeted) is selected separately for the GrIS and AIS. These models show the best agreement with present-day observations while maximizing the diversity of future projections. The authors show that CMIP5 models performing the best differ in Greenland and Antarctica, and that they do not represent atmospheric and oceanic processes equally well.

This is a sound, very well written study that is highly relevant for the Cryosphere community. The AOGCMs selected here will be used to force ice sheet models participating in the ISMIP6 project. Using the outputs of the best performing AOGCMs as forcing will prove essential to better project the mass balance of the GrIS and AIS in a future warming climate, and to improve estimates of their relative contribution to global sea level rise. I deem that the manuscript should be accepted for publication in the Cryosphere after applying some **minor revisions**. The authors can find my comments hereunder.

**General comments:**

1) The conclusion section should be reformulated to stress the main results of the study, i.e. purpose of the inter-comparison exercise, which climate models have been selected to force the ice sheet models, some perspective and future work based on e.g. CMIP6 models. The current conclusion section should better be moved to the discussion section. In addition, reference to Tables 2 and 3 appear for the first time in the conclusion section, while they should better be discussed at the end of Section 3.3 (Table 2) and Section 4.3 (Table 3).

2) The authors refer multiple times to forthcoming papers that are currently in preparation. I would strongly advise to remove those references or better use a personal communication statement as at e.g. **L61**, **L87**, **L144**, **L254-255**.

3) The authors should define the acronyms (e.g. ta850, prw, …) used for the evaluation metrics. These are currently not listed in the main manuscript making the interpretation of Figs. 2, 4, 5, 7 and A1 difficult. This should be done at **L107-111** (AIS) and **L113-114** (GrIS). For clarity, sea surface temperature in summer and sea ice extent in winter could be better defined as sst[s] (instead of tos[s]) and sie[w] (instead of mwsie). For consistency, I also suggest to replace ∂prw[a] by $\Delta$prw[a] in the main text and figures (e.g. L164-168). In addition, at L165-166, the authors refer to winter sea ice concentration (i.e. fraction of a pixel covered by sea ice) as opposed to sea ice extent (i.e. integrated area of pixels with a sea ice fraction > 0.15). Please clarify which quantity is used in both cases.

**Point comments:**

**L19**: The authors should also refer to more recent studies such as Mouginot et al. (2019; GrIS) and Shepherd et al. (2018; AIS). See also additional references.

**L39**: Remove "AOGCM" since it is first defined at **L41**.

**L49**: Could the authors provide a reference here (i.e. after ice shelves)?

**L50**: Could the authors provide a reference?

**L53-54**: The authors should add "e.g." before "Noël et al., 2018" and "van Wessem et al., 2018". For instance, Langen et al. (2017) and Niwano et al. (2018) also show good agreement between HIRHAM5 and NHM-SMAP RCMs and in situ measurements over the GrIS.

**L107-109**: I strongly suggest: "850 hPa (ta850; average of […] precipitable water (prw), […] pressure (psl), temperature (sst[s]) and winter sea ice extent (sie[w]) […] jet strength (Jstr) and position (Jpos), […] maximum in annual mean 850 hPa zonal wind […]".

**L113-115**: I strongly suggest: "[…] 700 hPa (ta700; average […] at 500 hPa (zg500), inside the Modèle Atmosphérique Régional (MAR; Fettweis et al., 2017) […]".

**L115**: "do not significantly impact MAR results".

**L131**: "ORCA025". **L138**: Could the authors provide a reference here?

**L141-142**: "World Ocean Atlas (WOA; Locarnini and […] 2018 WOA data (Locarnini […]"

**L166 and L169**: Add "(ΔT)" after "ocean temperature".

**L173**: At L156, the authors refer to 7 metrics for Greenland, while "6" is stated at L173. Do the authors discard Δzg500 from the comparison between future climate projections? Please, clarify.

**L182**: Add "(Fig. 2a)" after 0.13 and "a" after "Figure 2". **L183**: Add "(blue)", "(brown)" and "(yellow)" after "sub-surface ocean", "atmosphere" and "surface ocean". Same at **L240**: "(pink)", "(red)" and (light blue)"; at **L249** "(yellow)", **L254**: "(brown)" and "(dark blue)".

**L208**: I suggest: "We highlight the 3 core (red) and 3 targeted (yellow) AOGCMs selected in […]" and remove **L218-220**: "In Fig. 3b […] Amundsen region".

**L236**: Top 3 (core models). **L248**: Top 6 (targeted models) and same at **L314** and **L328**.

**L242**: Add "(dashed)" after "median".

**L250**: Do the authors mean "showing similar median projections under RCP8.5"? Please, clarify.

**L266**: The authors certainly mean "(Sections 4.1 and 4.2) and ensemble selection (Section 4.3)".

**L303**: "highlighted in Fig. 6b".

**L307**: $R^2$ = 0.31 is a weak correlation. Please, clarify.

**L312**: "[…] show that RCMs outperform global climate models […]".

**L315**: What about EC-EARTH? No values are shown in e.g. Fig. 6. Could the authors elaborate?

**L317**: Add "Fig. 7a" after "median".

**Section 4.3.1**: The authors should refer to Figs. C3 and C4 here.

**L333**: For consistency, ΔT BB (instead of Baffin Bay) and "ACCESS1-3".

**L347**: Do the authors mean that a similar evaluation/model selection and ranking is not planned/possible using CMIP6 models? Or that the evaluation/selection of CMIP6 models was not performed in the current study? Please, clarify.

**L408**: "ACCESS1-3".

**Stylistic comments:**

**L12**: Maybe "limitations" instead of "constraints". **L17**: I suggest: "[…] most uncertain contributors to global sea-level rise over multidecadal to millennial timescales.". **L35**: I suggest: "[…] and oceanic forcing contribution to the mass balance of both ice sheets vary greatly, and depend on […]".

**L44**: Maybe "converting" instead of "translating". **L45-46**: I suggest: "[…] resolution that is too coarse […] gradients impacting the surface climate of the ice sheets […]". **L55**: Maybe "unable" instead of "challenged". **L73**: I suggest: "[…] some of the limitations of the selection procedure, and discuss […]".

**L111**: "[…] (in m s$^{-1}$), compared to time-slice […]". **L203**: Maybe add "(core)" after "top 3 models".

**L206**: "multi-model". **L225**: "but this region is projected to warm moderately […]".

**L235**: Remove "to choose from". **L238**: I suggest: "The correction is robust and removes […] a time and changes the weight […]". **L257**: Maybe "large number" instead of "high number".

**L299**: Remove one of the double "an". **L359**: I suggest: "key processes for projections may still be missing.". **L366**: "models are assessed". **L367**: "evidenced in our analysis.". **L369**: "Concerning independence". **L387**: "e.g. Agosta et al., 2015" and "Meijers et al., 2012" before "Sallée et al., 2013".

**L392**: "their results differ from the current study […] ocean-driven basal melt".

**L397**: "the different model performance". **L401**: "ice flux of the different ice sheets". **L403**: Maybe "reasonable" instead of "feasible". **L404**: "RCP scenarios, …, parameters setting, …".

**L415**: I suggest: "better or project climate warming at different rates.". **L423**: "We refer readers interested in the […] simulations to Slater et al. (2019)."

**Figures:**

**Fig. 1:** Does the grey mask in Fig. 1a also represent the AIS regions above 2000 m a.s.l. as in Fig. 4c? Please clarify in the caption. **L107**: Does the blue rectangle in Figs. 1a,c represent the integration domain of MAR? If so, "standard lateral boundaries of MAR (REFs for AIS and GrIS)". **L112**: State the time period used for the "reference historical climatology". In addition, move the titles of Figs. 1a and c upward so that they do not overlap with the figures.

**Fig. 2:** For consistency replace legend items "surf. bias" and "ocean bias" by "surface ocean" and "sub-surface ocean". What do the vertical bars in Fig. 2a represent? Please, clarify. In Fig. 2b, add "ΔT" before DML, Amery, Totten, … The authors should also explicitly state that the horizontal dashed line represents the median of the models. Add "(core)" after top3 and "(targeted)" after top6.

**Fig. 5:** Same comments as for Fig. 2. The blue legend item in Fig. 5a should be "sub-surface ocean". In Fig. 5b, add "ΔT" before SPG, Baffin Bay … In Fig. 5c, "zg500" instead of "zg550hPa".

**Fig. 4:** For better contrast, I strongly suggest using a red line instead of the purple one for MIROC-ESM-CHEM. **Fig. 7:** Use a red line instead of the purple one for MIROC5.

**Fig. 6**: The figure titles and labels are too small and almost unreadable. Please, enlarge.

**Figs. B1 and B2**: As for Figs. 3 and 6, highlight the 3 core models in red and the 3 targeted models in yellow.

**Additional references:**

1) Shepherd et al. (2018): https://www.nature.com/articles/s41586-018-0179-y

2) Mouginot et al. (2019): https://www.pnas.org/content/116/19/9239

3) Langen et al. (2017): https://www.frontiersin.org/articles/10.3389/feart.2016.00110/full

4) Niwano et al. (2018): https://www.the-cryosphere.net/12/635/2018/

---

## Author Comment (AC1) · 21 Nov 2019

Response to reviewers:

*We thank the reviewers for their thoughtful reviews and are pleased that the manuscript was well received.*

*Both reviewers recommended to improve the conclusions section to highlight the main results of this study. We significantly increased the scope of the conclusions. They also both suggested minor corrections to improve the clarity of the manuscript, and we endeavored to implement these changes.*

*Reviewer #1 rightly pointed out the importance of the SAM and/or ASL in Antarctic regional variability. We thank them for noting that a comprehensive analysis of the SAM/ASL representation is beyond the scope of this study, but we agree that including a mention of this mode of variability adds to the manuscript. We therefore included it in the discussion section. Following their recommendation, we also fleshed out the appendices to summarize the findings from each of the appendix figures.*

*Reviewer #2 suggested changes to the variables notations, which we are happy to implement to improve the readability of the manuscript. They also shared concerns about referencing upcoming papers. We will update the references to (a) include the DOI of papers if applicable, (b) switch the references to personal communication if the relevant manuscripts are still unavailable.*

*Our responses to each of the reviewers comments are included in blue italic below.*

**Anonymous Referee #1**

Summary This article summarizes the selection of atmosphere-ocean coupled climate models (AOGCMs) to use for forcing for the stand-alone ice sheet simulations as part of the CMIP6 ice sheet intercomparison project (ISMIP6). The manuscript summarizes the methods used to select the models and recommends 6 models each for use with Antarctica and Greenland ice sheets. The models used in the selection process are those from CMIP5 AOGCMs (CMIP6 were insufficiently available for testing at the time of this analysis). Three "core" models are chosen for both Antarctica and Greenland based on their fidelity to observations during the satellite record period (1979-2005). Three more models ("targeted") were selected for use based on representation of a range of future atmosphere-ocean conditions from both the RCP2.6 and RCP8.5 emis- sions scenarios. This submission documents the selection criteria and subsequent of specific AOGCM selection for forcing of regional models for the ISMIP6. The work de- scribed is new and unique in that it uses both atmospheric and oceanic observations (rather than just atmospheric as in previous work) in the selection criteria. Antarctica and Greenland are treated separately, and with some overlapping and some unique variables as part of the evaluation of the AOGCMs.

The manuscript is well written, represents a significant scientific advance w.r.t. model selection for boundary conditions for ice sheet models. I recommend it be accepted for publication in The Cryosphere with minor revisions and technical corrections as follows:

*We thank Reviewer #1 for their thoughtful review and are pleased that the manuscript was well received. We thank Reviewer #1 for their suggested technical corrections and address them below.*

Minor revisions: Much of the regional variability in Antarctica is related to the zonal asymmetry in the Southern Annular Mode (SAM; or likewise the depth, location, and seasonal migration of the Amundsen Sea Low, ASL). Some models do a better job than others at capturing this – which is different than the metrics of zonal jet location and strength. There are many atmospheric and oceanic metrics used to select the model criteria in this submission, although none directly measure whether or not the models capture asymmetric nature of the SAM (although the combination of oceanic and atmospheric metrics used may indeed capture it indirectly). A full analysis of this (whether or not models capture this asymmetry, not to mention how, exactly, to measure if the models do) is beyond the scope of this paper. I do feel, however, that some mention is worthwhile – do you believe your metrics indeed capture this even if indirectly? Or do you think some of the regional biases might be due to a particular model's lack of an ASL? A model's fidelity or lack thereof to ASL could help explain some of the regional discrepancies in projected changes as well. (e.g. M. Holland, L. Landrum, Y. Kostov and J. Marshall, 2016, Sensitivity of Antarctic sea ice to the Southern Annual Mode in coupled climate models, Clim. Dyn., DOI 10.1007/s00382-016-3424-9; J. T. M. Lenaerts, J. Fyke, B. Medley. The signature of ozone depletion in recent Antarctic pre- cipitation change: a study with the Community Earth System Model, 2018. Geophys. Res. Lett., 45, 23, https://doi.org/10.1029/2018GL078608)

*We agree that the role of the SAM and/or ASL in regional variability is worth mentioning. We will include this point (and the suggested references) in the discussion section of the final manuscript.*

A couple sentences summarizing the figures/main point for each appendix would
be C2
helpful (have one sentence for Appendix C, none for A,
B).

*We added text to the appendices to summarize the main points of each appendix.*

Conclusions? Please
finish!

*We added text to flesh out the conclusions and highlight the goals and main findings of the study.*

Technical corrections (I can't figure out how to cut and paste Greek chi here so I write [chi]): Line 147: "historical metrics [chi] described above" but chi is not defined above. I believe [chi] in this case is the RMSE from the observations for each given variable – state this

*We modified this line to remove ambiguity: the historical metrics are described above, although chi is only used in the equation below.*

Lines 315-322: Section 4.3 Top 3 (Greenland) Last sentence in first paragraph ("model 1, model2") sounds like a placemarker – eliminate or re-write

*The placemaker "model1, model2" is intentional here. It is to highlight that the MIROC5 model was strategically chosen (and imposed), while the other two models were selected among the ensemble through our selection algorithm. The selection of model1 and model2 is explained in the following sentence.*

Lines 435-455 Check figure numbers. Mismatch between titles (in bold) and descrip- tions below (e.g. lines 439: "C2 Robustness of Antarctic . . .." Followed in line 440 by "Table C3 lists the . . .."

*We agree that the numbering of the appendices was confusing (section C1 applied to Tables C1 and C2, section C2 applied to Tables C3 and C4...). We adjusted the numbering to avoid confusion: the sections in appendix C are unnumbered, while the tables are numbered as C.1, C.2, etc. to be consistent with the appendix figure numbering system.*

Figure 1. Regional oceanic boundaries (and some of the text over the map of the continent) for Antarctica are difficult to see (very difficult in the printed version – better on the screen) – recommend trying for different colors, or perhaps thicker outlines of the regions. The most difficult regional texts are "Weddell (WS)" followed by "Amery (AM) caption:"Greenland...inside the usual boundaries of MAR simulations" define MAR?

*We thank Reviewer #1 for their feedback. We will endeavor to improve the readability of Figure 1 in the revised manuscript. We adjusted the caption to avoid confusion.*

Figures 2, 3, 5, 6, A.1, A.2, B.1, B.2 The symbols denoting models that were in top3 and top6 ensembles are very difficult to see (and not stated in captions for A.1, A.2). Figures 3 and 6 highlight w different colors so perhaps not as important in these, however in the other figures these symbols need to be easier to spot – with color, or bold, or?

*We will add color highlights to these figures in the revised manuscript.*

Table C2. Rewrite caption...says "three top models" and give statistics for four models (which are the four that give the two top-three combos)...

*We agree that the wording of the Table title was confusing. We adjusted the title to reflect that the model considered are those included in the possible combinations making the top 3.*

**Anonymous Referee #2**

With the aim of selecting a set of global climate models that represent best the current and projected climate of the Greenland (GrIS) and Antarctic ice sheets (AIS) to force the ice sheet models of ISMIP6, the authors evaluate, compare and rank 33 CMIP5 AOGCMs using observational data (present-day) combined with various atmospheric and oceanic metrics (scenarios). As a result, an ensemble of six AOGCMs (i.e. three core and three targeted) is selected separately for the GrIS and AIS. These models show the best agreement with present-day observations while maximizing the diversity of future projections. The authors show that CMIP5 models performing the best differ in Greenland and Antarctica, and that they do not represent atmospheric and oceanic processes equally well.

This is a sound, very well written study that is highly relevant for the Cryosphere community. The AOGCMs selected here will be used to force ice sheet models participating in the ISMIP6 project. Using the outputs of the best performing AOGCMs as forcing will prove essential to better project the mass balance of the GrIS and AIS in a future warming climate, and to improve estimates of their relative contribution to global sea level rise. I deem that the manuscript should be accepted for publication in the Cryosphere after applying some **minor revisions**. The authors can find my comments hereunder.

*We thank Reviewer #2 for their thoughtful review and are pleased that the manuscript was well received. We thank Reviewer #2 for their comments and suggested technical corrections, which we address below.*

**General comments:**

1) The conclusion section should be reformulated to stress the main results of the study, i.e. purpose of the inter-comparison exercise, which climate models have been selected to force the ice sheet models, some perspective and future work based on e.g. CMIP6 models. The current conclusion section should better be moved to the discussion section. In addition, reference to Tables 2 and 3 appear for the first time in the conclusion section, while they should better be discussed at the end of Section 3.3 (Table 2) and Section 4.3 (Table 3).

*Following the reviewers' suggestion, we added text to flesh out the conclusions and highlight the goals and main findings of the study. We also refer to Table 2 and 3 in Sections 3.3 and 4.3, as suggested.*

2) The authors refer multiple times to forthcoming papers that are currently in preparation. I would strongly advise to remove those references or better use a personal communication statement as at e.g. **L61**, **L87**, **L144**, **L254-255**.

*We had hoped for these papers to be published, or under review (with a DOI) by the time of publication of this manuscript. If it looks unlikely by the time of the final revision, we will replace these references with the link to the existing Wiki and/or personal communication statements.*

3) The authors should define the acronyms (e.g. ta850, prw, ...) used for the evaluation metrics. These are currently not listed in the main manuscript making the interpretation of Figs. 2, 4, 5, 7 and A1 difficult. This should be done at **L107-111** (AIS) and **L113-114** (GrIS). For clarity, sea surface temperature in summer and sea ice extent in winter could be better defined as sst[s] (instead of tos[s]) and sie[w] (instead of mwsie). For consistency, I also suggest to replace ∂prw[a] by △prw[a] in the main text and figures (e.g. L164-168). In addition, at L165- 166, the authors refer to winter sea ice concentration (i.e. fraction of a pixel covered by sea ice) as opposed to sea ice extent (i.e. integrated area of pixels with a sea ice fraction > 0.15). Please clarify which quantity is used in both cases.

*We adjusted the acronyms to improve readability, as suggested by the reviewer. For ∂prw[a], we do not replace it by △ as the ∂ notation indicates a different projected change: ∂ is for the difference divided by the mean over historical period. The text was adjusted to clarify the meaning of this metric. We thank the reviewer for pointing it out.*
*With regard to sea ice concentration vs. sea ice extent, it was indeed a mistake, we only used sea-ice extent. We corrected it in the text.*

**Point comments:** **L19**: The authors should also refer to more recent studies such as Mouginot et al. (2019; GrIS) and Shepherd et al. (2018; AIS). See also additional references. *We included these references as suggested.*

**L39**: Remove "AOGCM" since it is first defined at **L41**. *Adjusted as suggested.*

**L49**: Could the authors provide a reference here (i.e. after ice shelves)? *We will add a relevant reference in the revised manuscript.*

**L50**: Could the authors provide a reference? *We will add a relevant reference in the revised manuscript.*

**L53-54**: The authors should add "e.g." before "Noël et al., 2018" and "van Wessem et al., 2018". *Adjusted as suggested.* For instance, Langen et al. (2017) and Niwano et al. (2018) also show good agreement between HIRHAM5 and NHM-SMAP RCMs and in situ measurements over the GrIS.

**L107-109**: I strongly suggest: "850 hPa (ta850; average of [...] precipitable water (prw), [...] pressure (psl), temperature (sst[s]) and winter sea ice extent (sie[w]) [...] jet strength (Jstr) and position (Jpos), [...] maximum in annual mean 850 hPa zonal wind [...]". *Adjusted as suggested.*

**L113-115**: I strongly suggest: "[...] 700 hPa (ta700; average [...] at 500 hPa (zg500), inside the Modèle Atmosphérique Régional (MAR; Fettweis et al., 2017) [...]". *Adjusted as suggested.*

**L115**: "do not significantly impact MAR results". *Adjusted as recommended.*

**L131**: "ORCA025". *Adjusted as recommended.*

**L138**: Could the authors provide a reference here?
**L141-142**: "World Ocean Atlas (WOA; Locarnini and [...] 2018 WOA data (Locarnini [...]" *Adjusted as suggested.*
**L166 and L169**: Add "($\triangle$T)" after "ocean temperature". *Adjusted as suggested.*
**L173**: At L156, the authors refer to 7 metrics for Greenland, while "6" is stated at L173. Do the authors discard $\triangle$zg500 from the comparison between future climate projections? Please, clarify. **L182**: Add "(Fig. 2a)" after 0.13 and "a" after "Figure 2". *Adjusted as suggested.*
**L183**: Add "(blue)", "(brown)" and "(yellow)" after "sub-surface ocean", "atmosphere" and "surface ocean". Same at **L240**: "(pink)", "(red)" and (light blue)"; at **L249** "(yellow)", **L254**: "(brown)" and "(dark blue)". *Adjusted as suggested.*

**L208**: I suggest: "We highlight the 3 core (red) and 3 targeted (yellow) AOGCMs selected in [...]" and remove **L218-220**: "In Fig. 3b [...] Amundsen region". *Adjusted as suggested.*
**L236**: Top 3 (core models). **L248**: Top 6 (targeted models) and same at **L314** and **L328**. *We adjusted the title as follows:*
**L242**: Add "(dashed)" after "median". *Adjusted as suggested.*
**L250**: Do the authors mean "showing similar median projections under RCP8.5"? Please, clarify. *We reformulated the text to clarify the meaning.*
**L266**: The authors certainly mean "(Sections 4.1 and 4.2) and ensemble selection (Section 4.3)". **L303**: "highlighted in Fig. 6b". *Adjusted as suggested.*
**L307**: $R^2$ = 0.31 is a weak correlation. Please, clarify. *A correlation with $R^2$= 0.31 is still a moderate correlation (R > 50%), while we would consider $R^2$<= 0.25 to be a weak correlation.* **L312**: "[...] show that RCMs outperform global climate models [...]". *Adjusted as suggested.*
**L315**: What about EC-EARTH? No values are shown in e.g. Fig. 6. Could the authors elaborate? *At the time of the analysis, EC-EARTH future projections were not available for ocean data. We adjusted the wording to reflect that EC-EARTH was also disqualified due to data unavailability.*
**L317**: Add "Fig. 7a" after "median". *Adjusted as suggested.*
**Section 4.3.1**: The authors should refer to Figs. C3 and C4 here. *Adjusted as suggested.*
**L333**: For consistency, $\triangle$T BB (instead of Baffin Bay) and "ACCESS1-3". *Adjusted as suggested.*
**L347**: Do the authors mean that a similar evaluation/model selection and ranking is not planned/possible using CMIP6 models? Or that the evaluation/selection of CMIP6 models was not performed in the current study? Please, clarify. *We reformulated the text to clarify the meaning.*
**L408**: "ACCESS1-3". *Adjusted as suggested.*

**Stylistic comments:**

*We adjusted all the stylistic comments as suggested. We thank Reviewer #2 for their detailed reading, and their effort to improve the readability of the manuscript.*

**L12**: Maybe "limitations" instead of "constraints". **L17**: I suggest: "[...] most uncertain contributors to global sea-level rise over multidecadal to millennial timescales.". **L35**: I suggest: "[...] and oceanic forcing contribution to the mass balance of both ice sheets vary greatly, and depend on [...]". **L44**: Maybe "converting" instead of "translating". **L45-46**: I suggest: "[...] resolution that is too coarse [...] gradients impacting the

surface climate of the ice sheets [...]". **L55**: Maybe "unable" instead of "challenged". **L73**: I suggest: "[...] some of the limitations of the selection procedure, and discuss [...]". **L111**: "[...] (in m s⁻¹), compared to time-slice [...]". **L203**: Maybe add "(core)" after "top 3 models". **L206**: "multi-model". **L225**: "but this region is projected to warm moderately [...]". **L235**: Remove "to choose from". **L238**: I suggest: "The correction is robust and removes [...] a time and changes the weight [...]". **L257**: Maybe "large number" instead of "high number". **L299**: Remove one of the double "an". **L359**: I suggest: "key processes for projections may still be missing.". **L366**: "models are assessed". **L367**: "evidenced in our analysis.". **L369**: "Concerning independence". **L387**: "e.g. Agosta et al., 2015" and "Meijers et al., 2012" before "Sallée et al., 2013". **L392**: "their results differ from the current study [...] ocean-driven basal melt". **L397**: "the different model performance". **L401**: "ice flux of the different ice sheets". **L403**: Maybe "reasonable" instead of "feasible". **L404**: "RCP scenarios, ..., parameters setting, ...". **L415**: I suggest: "better or project climate warming at different rates.". **L423**: "We refer readers interested in the [...] simulations to Slater et al. (2019)."

**Figures:**

**Fig. 1:** Does the grey mask in Fig. 1a also represent the AIS regions above 2000 m a.s.l. as in Fig. 4c? Please clarify in the caption.

*No, the grey mask is over land for Antarctica, as stated in the caption ("For Antarctic atmosphere and surface ocean metrics, we considered the domain south of 40°S over ocean (color shading)")*

**L107**: Does the blue rectangle in Figs. 1a,c represent the integration domain of MAR? If so, "standard lateral boundaries of MAR (REFs for AIS and GrIS)". **L112**: State the time period used for the "reference historical climatology". In addition, move the titles of Figs. 1a and c upward so that they do not overlap with the figures. **Fig. 2:** For consistency replace legend items "surf. bias" and "ocean bias" by "surface ocean" and "sub-surface ocean".

What do the vertical bars in Fig. 2a represent? Please, clarify.

*The (light gray) vertical lines were added to increase the readability the figure: i.e. they link the various metrics of a given model so that we can easily compare the biases of one model.*

In Fig. 2b, add "△T" before DML, Amery, Totten, ... The authors should also explicitly state that the horizontal dashed line represents the median of the models. Add "(core)" after top3 and "(targeted)" after top6.

*In this manuscript, we reserve "△" as an indicator of the difference between end of 21st C and end of 20th century conditions. To avoid confusion, we do not use △ for the historical bias metrics. We adjusted the other points as suggested.*

**Fig. 5:** Same comments as for Fig. 2. The blue legend item in Fig. 5a should be "sub-surface ocean". In Fig. 5b, add "△T" before SPG, Baffin Bay ... In Fig. 5c, "zg500" instead of "zg550hPa".

*We thank the reviewer for noticing these details, we adjusted most of them as suggested. We decided to reserve the use of "△" to future changes (see comment above).*

**Fig. 4:** For better contrast, I strongly suggest using a red line instead of the purple one for MIROC- ESM-CHEM. **Fig. 7:** Use a red line instead of the purple one for MIROC5.

*We thank the reviewer for this suggestion. We adjusted the color of the pink line to increase contrast, as we prefer to avoid using red (already used for the "top 3" selection in other figures).*

**Fig. 6**: The figure titles and labels are too small and almost unreadable. Please, enlarge. *Adjusted as suggested.*

**Figs. B1 and B2**: As for Figs. 3 and 6, highlight the 3 core models in red and the 3 targeted models in yellow. *We will adjust the colors in the final manuscript.*

**Additional references:** 1) Shepherd et al. (2018): https://www.nature.com/articles/s41586-018-0179-y 2) Mouginot et al. (2019): https://www.pnas.org/content/116/19/9239 3) Langen et al. (2017): https://www.frontiersin.org/articles/10.3389/feart.2016.00110/full 4) Niwano et al. (2018): https://www.the-cryosphere.net/12/635/2018/